# Truncated proposals for scalable and hassle-free simulation-based inference

**Michael Deistler**
University of Tübingen
michael.deistler@uni-tuebingen.de

**Pedro J Gonçalves**\*
University of Tübingen
pedro.goncalves@uni-tuebingen.de

**Jakob H Macke**\*
University of Tübingen
Max Planck Institute for Intelligent Systems
jakob.macke@uni-tuebingen.de

## Abstract

Simulation-based inference (SBI) solves statistical inverse problems by repeatedly running a stochastic simulator and inferring posterior distributions from model-simulations. To improve simulation efficiency, several inference methods take a sequential approach and iteratively adapt the proposal distributions from which model simulations are generated. However, many of these sequential methods are difficult to use in practice, both because the resulting optimisation problems can be challenging and efficient diagnostic tools are lacking. To overcome these issues, we present Truncated Sequential Neural Posterior Estimation (TSNPE). TSNPE performs sequential inference with truncated proposals, sidestepping the optimisation issues of alternative approaches. In addition, TSNPE allows to efficiently perform coverage tests that can scale to complex models with many parameters. We demonstrate that TSNPE performs on par with previous methods on established benchmark tasks. We then apply TSNPE to two challenging problems from neuroscience and show that TSNPE can successfully obtain the posterior distributions, whereas previous methods fail. Overall, our results demonstrate that TSNPE is an efficient, accurate, and robust inference method that can scale to challenging scientific models.

## 1 Introduction

Computational models are an important tool to understand physical processes underlying empirically observed phenomena. These models, often implemented as numerical *simulators*, incorporate mechanistic knowledge about the physical process underlying data generation, and thereby provide an interpretable model of empirical observations. In many cases, several parameters of the simulator have to be inferred from data, e.g., with Bayesian inference. However, performing Bayesian inference in these models can be difficult: Running the simulator may be computationally expensive, evaluating the likelihood-function might be computationally infeasible, and the model might not be differentiable. In order to overcome these limitations, Approximate Bayesian Computation (ABC) methods [Beaumont et al., 2002, 2009], synthetic likelihood approaches [Wood, 2010], and neural network-based methods [e.g., Papamakarios and Murray, 2016, Hermans et al., 2020, Thomas et al., 2022] have been developed.

---

\*Equal contribution

36th Conference on Neural Information Processing Systems (NeurIPS 2022).

A subset of neural network-based methods, known as neural posterior estimation (NPE) [Papamakarios and Murray, 2016, Lueckmann et al., 2017, Greenberg et al., 2019], train a neural density estimator on simulated data such that the density estimator directly approximates the posterior. Unlike other methods, NPE does not require any further Markov-chain Monte-Carlo (MCMC) or variational inference (VI) steps. As it provides an *amortized* approximation of the posterior, which can be used to quickly evaluate and sample the approximate posterior for any observation, NPE allows the application in time-critical and high-throughput inference scenarios [Gonçalves et al., 2020, Radev et al., 2020, Dax et al., 2021], and fast application of diagnostic methods which require posterior samples for many different observations [Cook et al., 2006, Talts et al., 2018]. In addition, unlike methods targeting the likelihood (e.g., neural likelihood estimation, NLE [Papamakarios et al., 2019, Lueckmann et al., 2019]), NPE can learn summary statistics from data and it can use equivariances in the simulations to improve the quality of inference [Dax et al., 2021, 2022].

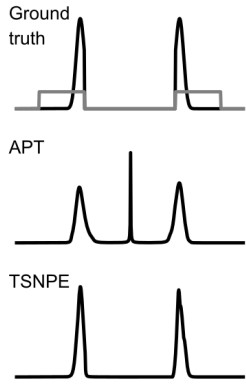

Figure 1: **APT vs TSNPE.** Top: Prior (gray) and true posterior (black). APT matches true posterior within the prior bounds but 'leaks' into region without prior support. TSNPE (ours) matches true posterior.

If inference is performed for a particular observation $\mathbf{x}_o$, sampling efficiency of NPE can be improved with *sequential* training schemes: Instead of drawing parameters from the prior distribution, they are drawn adaptively from a proposal (e.g., a posterior estimate obtained with NPE) in order to optimize the posterior accuracy for a particular $\mathbf{x}_o$. These procedures are called Sequential Neural Posterior Estimation (SNPE) [Papamakarios and Murray, 2016, Lueckmann et al., 2017, Greenberg et al., 2019] and have been reported to be more simulation-efficient than training the neural network only on parameters sampled from the prior, across a set of benchmark tasks [Lueckmann et al., 2021].

Despite the potential to improve simulation-efficiency, two limitations have impeded a more widespread adoption of SNPE by practitioners: First, the sequential scheme of SNPE can be unstable. SNPE requires a modification of the loss function compared to NPE, which suffers from issues that can limit its effectiveness on (or even prevent their application to) complex problems (see Sec. 2). Second, several commonly used diagnostic tools for SBI [Talts et al., 2018, Miller et al., 2021, Hermans et al., 2021] rely on performing inference across multiple observations. In SNPE (in contrast to NPE), this requires generating new simulations and network retraining for each observation, which often prohibits the use of such diagnostic tools [Lueckmann et al., 2021, Hermans et al., 2021].

Here, we introduce Truncated Sequential Neural Posterior Estimation (TSNPE) to overcome these limitations. TSNPE follows the SNPE formalism, but uses a proposal which is a *truncated* version of the prior: TSNPE draws simulations from the prior, but rejects them *before simulation* if they lie outside of the support of the approximate posterior. Thus, the proposal is (within its support) proportional to the prior, which allows us to train the neural network with maximum-likelihood in every round and, therefore, sidesteps the instabilities (and hence 'hassle') of previous SNPE methods. Our use of truncated proposals is strongly inspired by Blum and François [2010] and Miller et al. [2020, 2021], who proposed truncated proposals respectively for regression-adjustment approaches in ABC and for neural ratio estimation (see Discussion). Unlike methods based on likelihood(-ratio)-estimation [Miller et al., 2021, Hermans et al., 2021], TSNPE allows direct sampling and density evaluation of the approximate posterior, and thus permits computing expected coverage of the full posterior quickly (without MCMC) and at every iteration of the algorithm, thus allowing to diagnose failures of the method even for high-dimensional parameter spaces (we term this 'simulation-based coverage calibration' (SBCC), given its close connection with simulation-based calibration, SBC, Cook et al. [2006], Talts et al. [2018]).

We show that TSNPE is as efficient as the SNPE method 'Automatic Posterior Transformation' (APT, Greenberg et al. [2019]) on several established benchmark problems (Sec. 4.1). We then demonstrate that for two challenging neuroscience problems, TSNPE—but not APT—can robustly identify the posterior distributions (Sec. 4.2).

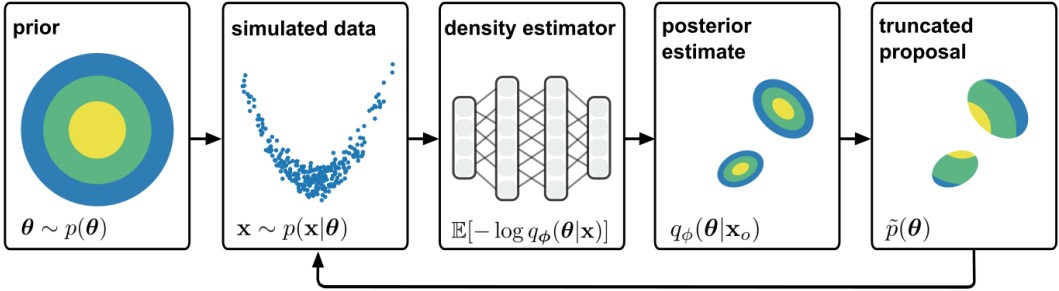

Figure 2: **Truncated Sequential Neural Posterior Estimation (TSNPE).** The method starts by sampling from the prior, running the simulator, and training a neural density estimator with maximum-likelihood to approximate the posterior. In subsequent rounds, parameters are sampled from the prior, but rejected if they lie outside of the support of the approximate posterior. With these proposals, the neural density estimator can be trained with maximum-likelihood in all rounds.

## 2 Background

In Neural Posterior Estimation (NPE), parameters are sampled from the prior $p(\boldsymbol{\theta})$ and simulated (i.e., $\mathbf{x}$ is sampled from $p(\mathbf{x}|\boldsymbol{\theta})$). Then, a neural density estimator $q_\phi(\boldsymbol{\theta}|\mathbf{x})$ (in our case a normalizing flow), with learnable parameters $\phi$, is trained to minimize the loss:

$$\min_\phi \mathcal{L} = \min_\phi \mathbb{E}_{\boldsymbol{\theta}\sim p(\boldsymbol{\theta})} \mathbb{E}_{\mathbf{x}\sim p(\mathbf{x}|\boldsymbol{\theta})}[-\log q_\phi(\boldsymbol{\theta}|\mathbf{x})],$$

which is minimized if and only if, for a sufficiently expressive density estimator, $q_\phi(\boldsymbol{\theta}|\mathbf{x}) = p(\boldsymbol{\theta}|\mathbf{x})$ for all $\mathbf{x} \in \text{supp}(p(\mathbf{x}))$ [Paige and Wood, 2016, Papamakarios and Murray, 2016]. Throughout this study, we refer to training with this loss function as maximum-likelihood training, although the neural density estimator targets the posterior directly.

Sequential Neural Posterior Estimation (SNPE) aims to infer the posterior distribution $p(\boldsymbol{\theta}|\mathbf{x}_o)$ for a particular observation $\mathbf{x}_o$. SNPE initially performs NPE and, thereby, obtains an initial estimate of the posterior distribution. It then samples parameters from a proposal $\tilde{p}(\boldsymbol{\theta})$, which is often chosen to be the previously obtained estimate of the posterior $\tilde{p}(\boldsymbol{\theta}) = q_\phi(\boldsymbol{\theta}|\mathbf{x}_o)$, and retrains the neural density estimator [Papamakarios and Murray, 2016]. This procedure can be repeated for several rounds.

Importantly, if parameters $\boldsymbol{\theta}$ are sampled from the proposal $\tilde{p}(\boldsymbol{\theta})$ rather than from the prior $p(\boldsymbol{\theta})$, the estimator $q_\phi(\boldsymbol{\theta}|\mathbf{x})$ that minimizes the maximum-likelihood loss function no longer converges to the true posterior. If one used the maximum-likelihood loss on data sampled from $\tilde{p}(\boldsymbol{\theta})$, i.e., $\mathcal{L} = \mathbb{E}_{\boldsymbol{\theta}\sim\tilde{p}(\boldsymbol{\theta})} \mathbb{E}_{\mathbf{x}\sim p(\mathbf{x}|\boldsymbol{\theta})}[-\log q_\phi(\boldsymbol{\theta}|\mathbf{x})]$, then $\mathcal{L}$ would be minimized by $q_\phi(\boldsymbol{\theta}|\mathbf{x}) \propto p(\boldsymbol{\theta}|\mathbf{x})\frac{\tilde{p}(\boldsymbol{\theta})}{p(\boldsymbol{\theta})}$, which is not the true posterior. Multiple schemes have been developed to overcome this [Papamakarios and Murray, 2016, Lueckmann et al., 2017]. The most recent of these methods, Automatic Posterior Transformation (APT, or SNPE-C, in its atomic version) [Greenberg et al., 2019, Durkan et al., 2020] employs a loss that aims to classify the parameter set that generated a particular data point among other parameter sets (details in Appendix Sec. 6.5).

While APT has been reported to significantly outperform previous methods, several studies have also described cases in which the approach exhibits performance issues: Both the original APT paper [Greenberg et al., 2019] and Durkan et al. [2020] reported that APT can show 'leakage' of posterior mass outside of bounded priors. We demonstrate this issue on a simple 1-dimensional simulator with bounded prior (Fig. 1, Appendix Fig. 7). The posterior estimated by APT is only required to match the true posterior density *within the support of the prior* (details in Appendix Sec. 6.5). Thus, after five rounds of APT, while the approximate posterior matches the true posterior within the bounds of the prior, a substantial fraction of posterior mass lies in regions with zero prior probability. In simple models, approximate posterior samples that lie outside of the prior bounds can be efficiently rejected. However, in models with high numbers of parameters, the rejection rate can become so large that drawing posterior samples which lie inside of the prior bounds is prohibitive. For example, Glöckler et al. [2022] reported a rejection rate of more than 99.9999% in a model with 31 parameters, thus requiring approximately one minute to draw a single posterior sample from within the prior bounds.

**Algorithm 1:** TSNPE

---

**Inputs:** prior $p(\boldsymbol{\theta})$, observation $\mathbf{x}_o$, simulations per round $N$, number of rounds $R$, $\epsilon$ that defines
the highest-probability region (HPR$_\epsilon$)
**Outputs:** Approximate posterior $q_\phi$.
**Initialize:** Proposal $\tilde{p}(\boldsymbol{\theta}) = p(\boldsymbol{\theta})$, dataset $\mathcal{X} = \{\}$
**for** $r \in [1, ..., R]$ **do**
    **for** $i \in [1, ..., N]$ **do**
        $\boldsymbol{\theta}_i \sim \tilde{p}(\boldsymbol{\theta})$
        simulate $\mathbf{x}_i \sim p(\mathbf{x}|\boldsymbol{\theta}_i)$
        add $(\boldsymbol{\theta}_i, \mathbf{x}_i)$ to $\mathcal{X}$
    $\boldsymbol{\phi}^* = \arg\min_{\boldsymbol{\phi}} -\frac{1}{N} \sum_{(\boldsymbol{\theta}_i, \mathbf{x}_i) \in \mathcal{X}} \log q_\phi(\boldsymbol{\theta}_i|\mathbf{x}_i)$
    Compute expected coverage($\tilde{p}(\boldsymbol{\theta}), q_\phi$) ;             `// see Alg. 2`
    $\tilde{p}(\boldsymbol{\theta}) \propto p(\boldsymbol{\theta}) \cdot \mathbb{1}_{\boldsymbol{\theta} \in \text{HPR}_\epsilon}$ ;                    `// see Alg. 3`

---

We overcome these limitations by using 'truncated' proposal distributions. This allows us to train with maximum-likelihood at every round, thereby sidestepping issues of previous SNPE methods.

## 3 Methodology

### 3.1 Truncated proposals for SNPE

Given a particular observation $\mathbf{x}_o$, we suggest to restrict the proposals $\tilde{p}(\boldsymbol{\theta})$ to be proportional to the prior $p(\boldsymbol{\theta})$ at least in the $1 - \epsilon$ highest-probability-region (HPR$_\epsilon$, the smallest region that contains $1 - \epsilon$ of the mass) of $p(\boldsymbol{\theta}|\mathbf{x}_o)$, i.e.

$$\tilde{p}(\boldsymbol{\theta}) \propto p(\boldsymbol{\theta}) \cdot \mathbb{1}_{\boldsymbol{\theta} \in \mathcal{M}}$$

with HPR$_\epsilon(p(\boldsymbol{\theta}|\mathbf{x}_o)) \subseteq \mathcal{M}$. Thus, $\tilde{p}(\boldsymbol{\theta})$ is a 'truncated' proposal. The key insight is that, when using such a proposal and $\epsilon = 0$, one can train $q_\phi(\boldsymbol{\theta}|\mathbf{x})$ with maximum likelihood:

$$\min_{\boldsymbol{\phi}} \mathcal{L} = \min_{\boldsymbol{\phi}} \mathbb{E}_{\boldsymbol{\theta} \sim \tilde{p}(\boldsymbol{\theta})} \mathbb{E}_{\mathbf{x} \sim p(\mathbf{x}|\boldsymbol{\theta})} [-\log q_\phi(\boldsymbol{\theta}|\mathbf{x})],$$

and $q_\phi(\boldsymbol{\theta}|\mathbf{x}_o)$ will still converge to $p(\boldsymbol{\theta}|\mathbf{x}_o)$ (Proof in Appendix Sec. 6.2).

We estimate $\mathcal{M}$ as the HPR$_\epsilon$ of the approximate posterior $\mathcal{M} = $ HPR$_\epsilon(q_\phi(\boldsymbol{\theta}|\mathbf{x}_o))$. Since the maximum-likelihood loss employed to train $q_\phi(\boldsymbol{\theta}|\mathbf{x})$ is support-covering, the HPR$_\epsilon$ of $q_\phi(\boldsymbol{\theta}|\mathbf{x}_o)$ tends to cover the HPR$_\epsilon$ of $p(\boldsymbol{\theta}|\mathbf{x}_o)$ [Bishop and Nasrabadi, 2006].

In order to obtain the HPR$_\epsilon$ of $q_\phi(\boldsymbol{\theta}|\mathbf{x}_o)$, we define a threshold $\boldsymbol{\tau}$ on the approximate posterior density $q_\phi(\boldsymbol{\theta}|\mathbf{x}_o)$. To do so, we use a normalizing flow as $q_\phi(\boldsymbol{\theta}|\mathbf{x})$, which allows for closed-form density evaluation and fast sampling. We then approximate the HPR$_\epsilon$ of $q_\phi(\boldsymbol{\theta}|\mathbf{x}_o)$ as

$$\text{HPR}_\epsilon(q_\phi(\boldsymbol{\theta}|\mathbf{x}_o)) \approx \mathbb{1}_{q_\phi(\boldsymbol{\theta}|\mathbf{x}_o) > \boldsymbol{\tau}}.$$

We chose $\boldsymbol{\tau}$ as the $\epsilon$-quantile of approximate posterior densities of samples from $q_\phi(\boldsymbol{\theta}|\mathbf{x}_o)$, and evaluated TSNPE for $\epsilon = 10^{-3}, 10^{-4}$, and $10^{-5}$. Values of $\epsilon > 0$ yield a proposal prior which has smaller support than the current estimate of the posterior, e.g., using $\epsilon = 10^{-3}$ neglects 0.1% of mass from the approximate-posterior support. Thus, this approach leads to errors in posterior estimation, e.g., to 'under-covered' posteriors (Appendix Sec. 6.10). However, empirically, the error induced by this truncation is negligible, as we will demonstrate on several benchmark tasks. We note that TSNPE can be trained on data pooled from all rounds (Appendix Sec. 6.2). TSNPE is summarized in Alg. 1 (Fig. 2).

### 3.2 Sampling from the truncated proposal

To generate training data for subsequent rounds, we have to draw samples from the truncated proposal $\tilde{p}(\boldsymbol{\theta})$, and here we explored rejection sampling and sampling importance resampling (SIR) [Rubin, 1988]. For rejection sampling, we sample the prior $\boldsymbol{\theta} \sim p(\boldsymbol{\theta})$ and accept samples only if their probability under the approximate posterior $q_\phi(\boldsymbol{\theta}|\mathbf{x})$ is above threshold $\boldsymbol{\tau}$.

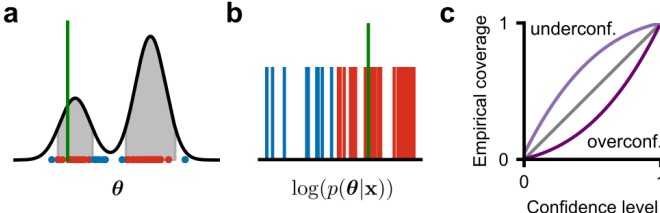

Figure 3: **Diagnostic tool.** (a) Parameter $\theta^*$ (green) lies within the 1-$\alpha$ confidence region (gray) of the estimated posterior. (b) $\log(p(\theta^*|x))$ is above the 1-$\alpha$ quantile of posterior samples. (c) 1-$\alpha$ versus empirical coverage, averaged over $\theta^*$.

This strategy samples from the truncated proposal exactly, but can fail if the rejection rate becomes too high. To deal with these situations, we used SIR. For each sample from the truncated proposal, SIR draws $K$ samples from the approximate posterior, computes weights $w_{i=1...K} = p(\theta_i)\mathbb{1}_{\theta_i \in \mathcal{M}}/q_\phi(\theta_i|\mathbf{x})$, normalizes $w_i$ such that they sum to one, draws from a categorical distribution with weights $s \sim \text{Categorical}(w_i)$, and selects the posterior sample with index $s$. SIR requires a fixed sampling budget of $K$ posterior samples per sample from the truncated proposal and returns exact samples from the truncated proposal for $K \to \infty$. Too low values of $K$ lead to too narrow proposals and posterior approximations. When run for a number of rounds, this behaviour reinforces itself and can lead to divergence of TSNPE (Appendix Fig. 13). We, thus, chose a high value $K = 1024$. In our experiments, we did not observe poor SIR performance, but we emphasise the importance of using tools to diagnose potential failures of TSNPE (see below) or SIR (e.g. by inspecting the effective sample size, Appendix Sec. 6.12). When SIR fails, methods such as nested sampling, adaptive multi-level splitting, or sequential Monte-Carlo sampling could be viable alternatives [Skilling, 2004, Cérou and Guyader, 2007, Doucet et al., 2001]. We discuss computational costs of rejection sampling and SIR in Appendix Sec. 6.11.

## 3.3 Coverage diagnostic

In order for the estimated posterior $q_\phi(\theta|\mathbf{x}_o)$ to converge to $p(\theta|\mathbf{x}_o)$, TSNPE requires $\text{supp}(p(\theta|\mathbf{x}_o)) \subseteq \text{HPR}_\epsilon(q_\phi(\theta|\mathbf{x}_o))$, i.e., the estimated posterior must be broader than the true posterior (proof in Appendix Sec. 6.2). In order to diagnose whether the posterior is, on average, sufficiently broad, we perform expected coverage tests as proposed in Dalmasso et al. [2020], Miller et al. [2021], Hermans et al. [2021].

As described in Dalmasso et al. [2020], Rozet et al. [2021] and illustrated in Fig. 3, the coverage of the approximate posterior can be computed as

$$1 - \alpha = \int q_\phi(\theta|\mathbf{x}^*)\mathbb{1}(q_\phi(\theta^*|\mathbf{x}^*) \geq q_\phi(\theta|\mathbf{x}^*))d\theta$$

where $\theta^*$ is sampled from the truncated proposal and $\mathbf{x}^*$ is the corresponding simulator output. In order to approximate this integral, one has to either evaluate the approximate posterior on a grid [Dalmasso et al., 2020, Hermans et al., 2021] or apply a Monte-Carlo average which includes repeatedly sampling (and evaluating) the (unnormalized) approximate posterior [Miller et al., 2021, Rozet et al., 2021]. The first option does not scale to high-dimensional spaces whereas the second is computationally expensive for methods estimating likelihood(-ratios) and, thus, require MCMC. In contrast, the TSNPE-posterior can be sampled from and evaluated in closed-form, leading to a computationally efficient and scalable diagnostic which can be run after every training round.

Expected coverage can be computed as an average of the coverage across multiple pairs $(\theta^*, \mathbf{x}^*)$ [Miller et al., 2021, Hermans et al., 2021] and should match the confidence level for all confidence levels $(1 - \alpha) \in [0, 1]$ (Fig. 3c). We term this procedure of computing the empirical coverage 'simulation-based coverage calibration' (SBCC), due to its close connection with SBC [Cook et al., 2006, Talts et al., 2018] (identical under certain conditions, Appendix Sec. 6.6). For TSNPE, it is important that the empirical expected coverage matches the confidence level for high confidence levels (i.e., for small $\alpha$), since overconfidence in these regions would indicate that ground-truth parameters $\theta^*$ are falsely excluded from the $\text{HPR}_{\epsilon=\alpha}$. SBCC is summarized in Appendix Alg. 2.

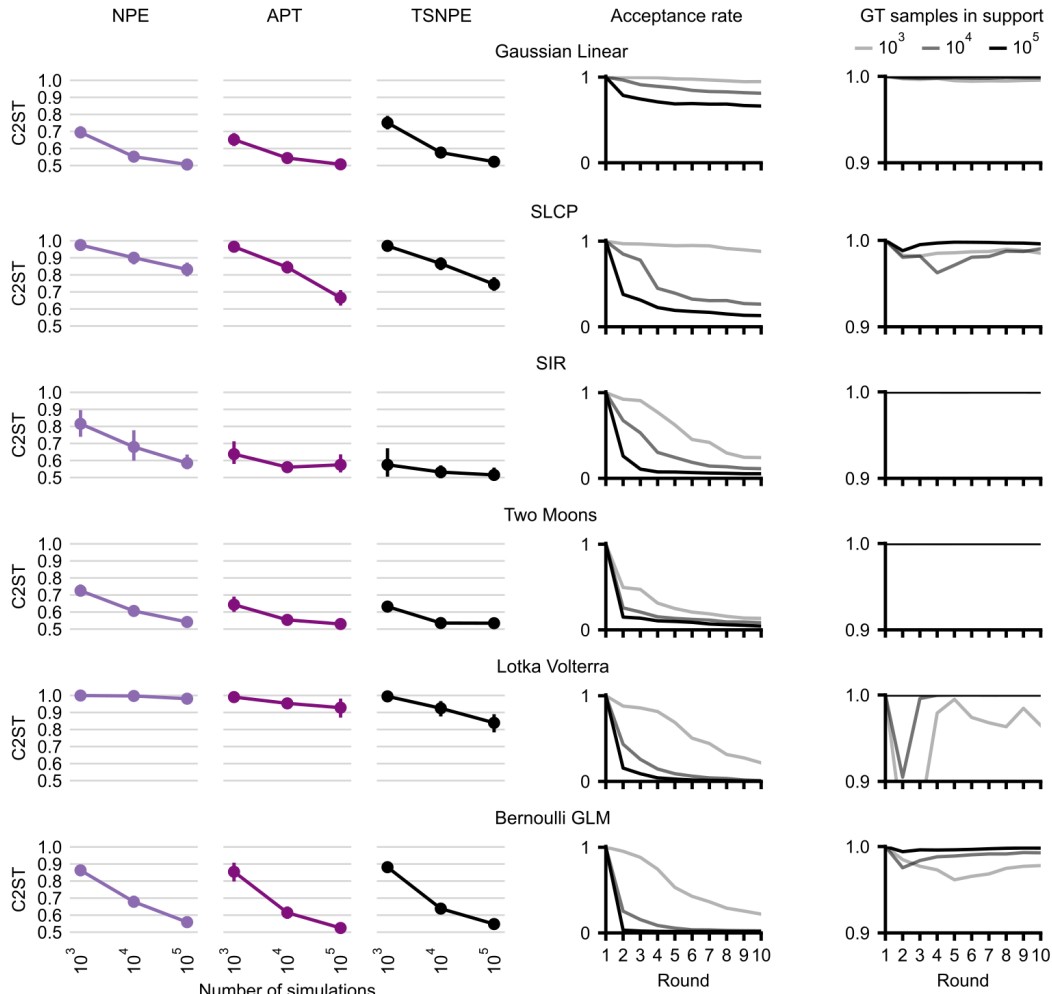

Figure 4: **Performance on six benchmark tasks.** Left three columns: classifier two-sample test accuracy (C2ST) of NPE (left), APT (middle), and TSNPE (right) for three simulation budgets. Forth column: Fraction of prior samples within the approximate-posterior $HPR_\epsilon$ in each round for each simulation budget. Fifth column: Fraction of true-posterior samples within the approximate-posterior $HPR_\epsilon$. TSNPE with $\epsilon = 10^{-4}$ and rejection sampling from truncated proposal.

# 4 Results

We evaluated TSNPE on several benchmark tasks and on two complex problems from neuroscience. We found that TSNPE performs as well as APT on the benchmark tasks and that it is robust to choices of $\epsilon$. In addition, we found that, in contrast with APT, TSNPE can successfully infer the posterior distribution for complex models with large numbers of parameters.

## 4.1 Performance on benchmark tasks

We compared TSNPE with NPE and APT on six benchmark tasks for which samples from the ground-truth posterior are available (see Appendix Sec. 6.9 for tasks) [Lueckmann et al., 2021]. We quantified the performance with a classifier two-sample test (C2ST), for which 0.5 indicates that the approximate posterior is identical to the ground-truth posterior, whereas 1.0 implies that the distributions can be completely separated by a classifier. Overall, APT and TSNPE perform similarly well and both outperform NPE (Fig. 4, left three columns). On two of the six tasks (Gaussian Linear and SLCP), APT has slightly better performance than TSNPE, whereas on two other tasks (SIR

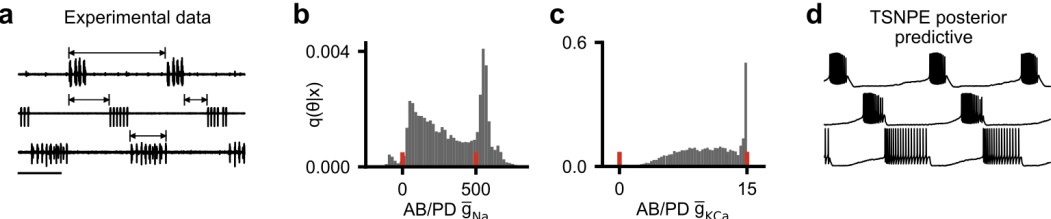

Figure 5: **Pyloric network inference.** (a) Data [Haddad and Marder, 2021]. (b) APT approximate posterior (1D-marginal) exhibits leakage (red: prior bounds). (c) APT approximate posterior when forcing the density estimator into constrained space. The spike at the upper prior bound is at odds with previously published posterior distributions [Gonçalves et al., 2020, Deistler et al., 2021, Glöckler et al., 2022] and produces poor predictive samples (Appendix Fig. 19). (d) TSNPE posterior predictive sample matches summary statistics of the experimental data.

and Lotka-Volterra), TSNPE outperforms APT. Overall, TSNPE and APT perform similarly well, demonstrating that TSNPE is competitive with previous methods on benchmark tasks.

In order to get insights into the improved performance of TSNPE as compared to NPE, we computed the fraction of prior samples that lie within the $\text{HPR}_\epsilon$ of the approximate posterior (Fig. 4, fourth column). In tasks with broad posteriors and few simulations, the $\text{HPR}_\epsilon$ is almost as wide as the prior and thus the performances of NPE and TSNPE are similar (e.g., SLCP with 1k simulations). In other tasks and with more simulations, the $\text{HPR}_\epsilon$ is much narrower than the prior, leading to an improvement in simulation efficiency (e.g., Lotka-Volterra with 100k simulations).

Finally, we evaluated whether the $\text{HPR}_\epsilon$ of the approximate posterior contains the support of the true posterior (Fig. 4, fifth column). We computed the fraction of true-posterior samples within the $\text{HPR}_\epsilon$ of the approximate posterior. For most tasks, fewer than 0.1% of samples were excluded, and the rate of erroneously rejected samples decreased as more simulations were used. In the Lotka-Volterra task with 1k and 10k simulations, many ground-truth samples were rejected and TSNPE performed poorly, but NPE and APT also failed to solve the task. Thus, while truncated proposals can potentially induce posterior biases, these only have a negligible effect on the performance of TSNPE. We note that TSNPE performance is qualitatively unaffected by the choice of $\epsilon \leq 10^{-4}$ and proposal sampling scheme (Appendix Fig. 8, Fig. 9, Fig.10). Applying truncated proposals to APT leads to equally good or worse performance than 'standard' APT, depending on the task (Appendix Fig. 14).

### 4.2 Efficient and robust inference in two complex neuroscience problems

Next, we evaluate the performance of TSNPE on two challenging neuroscience problems, where the competitive advantage of TSNPE is fully realized.

**Pyloric network** We applied TSNPE to a challenging real-world simulator from neuroscience: The pyloric network of the stomatogastric ganglion in the crab *Cancer Borealis* [Prinz et al., 2003, 2004]. The model has 31 parameters and simulates 3 voltage traces that we reduce to 18 summary statistics. The prior distribution is uniform within previously described parameter ranges [Prinz et al., 2004, Gonçalves et al., 2020]. We identify the posterior distribution given experimentally observed data [Haddad and Marder, 2021] (Fig. 5a) with APT and TSNPE (13 rounds, 30k simulations per round).

When applying APT 'out of the box' (from 'sbi' toolbox [Tejero-Cantero et al., 2020]), the rate of approximate-posterior samples within the prior bounds was 0.02% after the second round and 0.0000% after the third round (Fig. 5b), which rendered a fourth round too computationally expensive.

We attempted to overcome these issues by appending a transformation $T$ to the density estimator $q_\phi(\boldsymbol{\theta}|\mathbf{x})$ such that its support is constrained to match the support of the prior. In practice, we used a sigmoid transformation. While the resulting approximate posterior exhibited no leakage, this setup revealed another problem when running APT: In transformed (i.e., unbounded) space, the density estimator $q_\phi(\boldsymbol{\theta}|\mathbf{x})$ can put significant mass in regions outside of the training data. When forced into constrained space, these 'leaking' regions lead to spikes at the bounds of the parameter space (Fig. 5c, further details in Appendix Sec. 6.5; illustration, additional tests and full posterior in Appendix

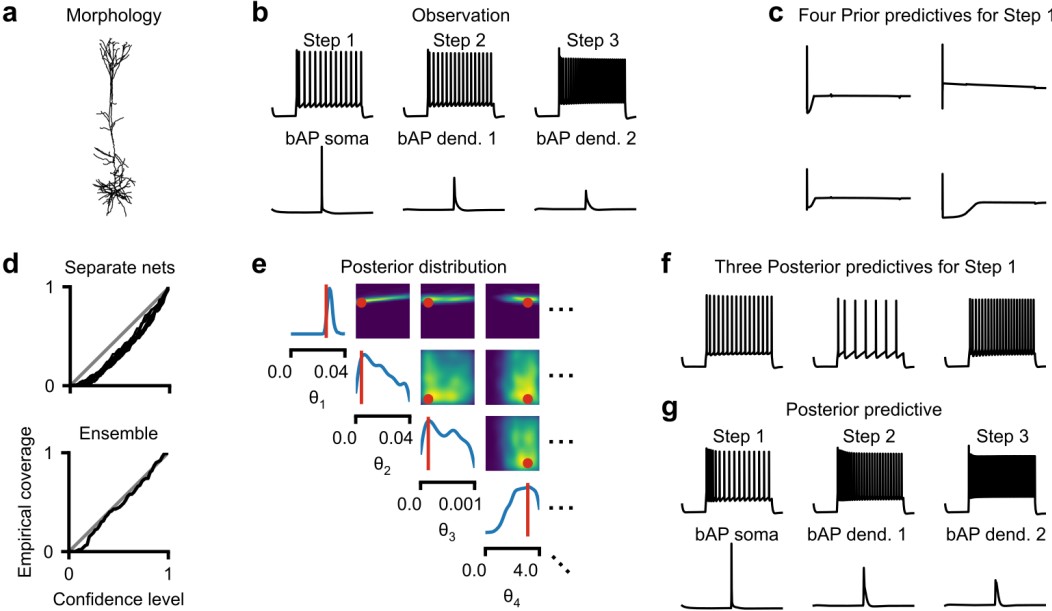

Figure 6: **TSNPE on L5PC.** (a) Cell morphology. (b) Observation. (c) Four prior samples for the Step 1 protocol. (d) Coverage for 1 and 10 neural nets. (e) Posterior. True parameter in red. (f) Three posterior predictives for the Step 1 protocol. (g) One posterior predictive for all protocols.

Figs. 17, 16 and 18). These spikes are at odds with previously published posterior distributions [Gonçalves et al., 2020, Deistler et al., 2021, Glöckler et al., 2022] and samples from these parameter regions do not produce good simulations (Appendix Fig. 19). This demonstrates that leakage occurs in APT even when the density estimator is forced into constrained space and that these issues lead to an incorrect posterior approximation as well as to poor predictive samples.

We applied TSNPE to this task for 13 rounds without any issue. The resulting posterior produces samples that closely match the observed data (Fig. 5d, more samples in Appendix Fig. 15, posterior distribution across all 31 parameters in Appendix Fig. 20). The obtained posterior is similar to previously published posteriors [Gonçalves et al., 2020, Deistler et al., 2021, Glöckler et al., 2022].

**Multicompartment model of a single neuron**    Finally, we turn to a landmark problem in neuroscience for which the posterior has not yet been identified: A morphologically detailed model of a thick-tufted layer 5 pyramidal cell (L5PC) from the neocortex [Ramaswamy et al., 2015, Markram et al., 2015, Van Geit et al., 2016]. The model describes the response of a neuron to current stimuli of different strengths. The model has approximately 7000 separate compartments which compose the anatomy of the cell (Fig. 6a). Each compartment has dynamics based on the Hodgkin-Huxley equations [Hodgkin and Huxley, 1952] and contains multiple ion channels (details in Van Geit et al. [2016]). The model has 20 free parameters which are the maximal channel conductances and time constants of the ion channels. The simulation consists of four separate simulations corresponding to experimental protocols which describe the voltage response to different stimuli. For the first three protocols (Step 1, Step 2, Step 3), each voltage response is characterized by 10 summary statistics. The fourth protocol models the back-propagation of the voltage response through the dendritic tree and is captured by 5 additional summary statistics (bAP soma, bAP dend. 1, bAP dend. 2). In total, the model produces 35 summary statistics, to which we add Gaussian noise with diagonal covariance matrix capturing the response variability of previously reported measurements [Hay et al., 2011].

Our goal is to infer the posterior distribution over 20 parameters given 35 summary statistics that were simulated—and thus have a known ground-truth parameter set— to resemble experimentally observed activity (Fig. 6b). The prior is a uniform distribution within previously established bounds [Van Geit et al., 2016]. A major difficulty in fitting this model is that a large fraction of prior samples generate summary statistics that are very different from the observed data: In particular, about 99.98% of prior predictives contain at least one summary statistic that is undefined, e.g., time to first spike is

undefined in the absence of spikes (Fig. 6c). When a summary statistic is undefined, we assign it a value that is substantially outside the range of the observed data (Appendix Sec. 6.14).

We ran TSNPE over six rounds (hyperparameters in Appendix Sec. 6.15). In each round, we ran 30k simulations, leading to a total of 180k simulations. After every round, we evaluated the expected coverage with SBCC. After the first round, the approximate posterior exhibited poor expected coverage (Fig. 6d top). Therefore, as suggested by Hermans et al. [2021], we used an ensemble of 10 neural density estimators to ensure that the approximate posterior is sufficiently broad (Fig. 6d bottom). Although the approximate posterior remains underconfident, the empirical expected coverage closely matches the confidence level for high-confidence levels, which is crucial for TSNPE (Sec. 3.3).

The TSNPE-posterior has several parameters with broad marginals, demonstrating that this model exhibits 'degeneracy', a widespread phenomenon in biological systems [Marder and Taylor, 2011] (Fig. 6e, Appendix Sec. 6.14 for parameter names). Other marginals are narrow, demonstrating that the model is sensitive to changes in these parameters. Posterior predictive samples closely match the observed data (Fig. 6f,g; more samples in Appendix Fig. 22). We emphasize that fitting such morphologically-detailed neuron models is a challenging and widespread problem in neuroscience, one for which commonly used methods [e.g., genetic algorithms, Druckmann et al., 2007, Van Geit et al., 2016] are often simulation-inefficient or do not estimate the full posterior distribution. We show promising results, suggesting that TSNPE could be applied to other complex single-neuron models.

In contrast, with 'out of the box' APT, none of the 10M approximate-posterior samples was within the prior bounds after the second round. This rendered a third round too computationally expensive. Overall, the results on the pyloric network model and on the multicompartment model demonstrate that TSNPE is an efficient and robust method that scales to complex and high-dimensional models that were inaccessible to the state-of-the-art method APT.

## 5 Discussion

We presented a new method to perform Bayesian inference in implicit models, which we call Truncated Sequential Neural Posterior Estimation (TSNPE). Like previous methods, TSNPE adaptively selects parameters to improve simulation-efficiency and allow posterior inference in complex models with many parameters. The key ingredient is that TSNPE samples parameters from a 'truncated' region of the prior, and thus overcomes instabilities of previous methods while maintaining simulation efficiency. In order to diagnose potential errors of TSNPE, we developed a coverage test that can be run quickly and at every round of TSNPE. TSNPE presents a new variant of SNPE which is at least as powerful as previous variants on benchmark tasks, but provides a powerful alternative which is able to solve inference problems on which the state-of-the-art method APT failed.

**Related work**   TSNPE differs from automatic posterior transformation (APT, SNPE-C) in its proposal and its loss function: TSNPE uses a truncated prior as proposal, while APT can flexibly use any proposal, which, e.g., allows for more sophisticated active learning rules [Lueckmann et al., 2019, Järvenpää et al., 2019]. However, APT's flexibility requires a modification of NPE loss function, which can be an impediment to its usage in practice: First, the modification can lead to 'leakage', which can make it prohibitive to draw samples within prior bounds. Second, APT loss requires an explicit prior and thus, cannot be applied to models in which the prior can only be sampled [Ramesh et al., 2022]. Third, current formulations of APT cannot discard parameters leading to invalid simulations as the posterior mass would 'leak' into parameter regions which only produce invalid simulations (Appendix Sec. 6.5). It might be possible that these issues are resolved using a modified formulation of APT, e.g., by combining its atomic loss with additional loss terms, or preventing leakage by penalizing 'bad' parameters [Greenberg et al., 2019]. In cases in which leakage prevents application of APT (in particular, in high-dimensional problems), TSNPE provides an alternative.

Our method is inspired by previous work that introduced a mechanism to post-hoc correct samples obtained by an Approximate Bayesian Computation (ABC) algorithm [Blum and François, 2010], i.e., 'regression adjustment ABC'. Their method draws samples from a truncated region of the prior to avoid correction terms, but estimates the posterior density with ABC samples—rather than using a flexible neural density estimator—and estimates the support by training a dedicated support-vector machine. In addition, the method runs a single round of truncation and retraining, whereas we demonstrate that TSNPE can be robustly applied across 10 rounds.

Truncated proposals have also been proposed for neural ratio estimation [Truncated Marginal Neural Ratio Estimation (TMNRE) Miller et al., 2021]: TMNRE uses truncated proposals to efficiently infer selected posterior marginals while being amortised around the observation, allowing to test the coverage properties of the selected marginals, e.g., with SBC [Cook et al., 2006, Talts et al., 2018]. In addition, truncating based on the marginals allows TMRNE to sample from the truncated proposal without rejection or SIR sampling. In contrast, TSNPE aims at efficiently inferring the full posterior distribution by proposals that avoid the correction of SNPE loss function. Truncating the proposal based on the full posterior rather than on the marginals can lead to drastically narrower proposals: E.g., on the pyloric network problem, truncation based on posterior marginals rejects 20% of prior samples versus 99.94% rejection based on the posterior joint. In addition, while TMNRE uses the expected coverage to test the consistency of the posterior marginals, TSNPE can test the expected coverage of the full posterior distribution.

**Possible failure modes**   The main failure mode of TSNPE will occur if the truncated proposal excludes significant portions of density mass of the true posterior (e.g., if the estimate misses posterior modes). In these cases, the learned approximate posterior will put systematically too little mass in the excluded regions. We recommend the use of diagnostic tools such as SBCC to identify such failures [Cook et al., 2006, Miller et al., 2021, Hermans et al., 2021, Rozet et al., 2021].

In addition, if the true posterior has unbounded support, any finite values of $\epsilon > 0$ will lead to a biased approximate posterior which puts too little weight in the posterior tails. In that case, and when running TSNPE across many rounds, the errors from each individual round could accumulate. Although we did not observe this bias to significantly affect the algorithm performance on several benchmark tasks, we cannot exclude the possibility of a substantial performance degradation when running TSNPE for a larger number of rounds ($\gg$10).

Finally, unlike SNPE methods that use the previous estimate of the posterior as the proposal distribution, our method requires a scheme to sample from a truncated proposal. If the sampling scheme is inaccurate (i.e., if it does not produce a proposal distribution that is proportional to the prior within the truncated region), the results of TSNPE will be biased. To avoid this, we recommend using rejection sampling by default and using SIR or sequential Monte-Carlo methods only if rejection sampling is too computationally expensive. For SIR, it is important to use a large oversampling factor $K$ (e.g., $K = 1024$) and use diagnostic tools such as effective sample size (Appendix Sec. 6.12, Fig. 13).

**Simulation-based coverage calibration**   In order to diagnose whether the approximate posterior is broader than the true posterior, we applied SBCC, a coverage test for TSNPE [Cook et al., 2006, Rozet et al., 2021]. SBCC evaluates the expected coverage of the approximate posterior without evaluating it on a grid [Dalmasso et al., 2020, Hermans et al., 2021] and, unlike diagnostic tools for methods based on learning the likelihood(-ratio), does not require MCMC runs for multiple observations $\mathbf{x}$ [Miller et al., 2021]. This allows SBCC to be run quickly and for models with many parameters. In addition, in contrast to diagnostic tools for likelihood-free inference with Approximate Bayesian Computation, SBCC does not require an additional step to estimate the density of approximate posterior samples [Prangle et al., 2014]. We note that, since SBCC is a variation of SBC [Cook et al., 2006, Talts et al., 2018], it only ensures that the $\text{HPR}_\epsilon$ is correct *on average* across observations, not for a particular observation. In principle, SBCC could be applied to other SNPE variants, although empirically the impact of arbitrary proposals on SBCC performance is currently unclear.

**Conclusion**   Overall, TSNPE combines the simulation-efficiency of sequential neural posterior estimation with the robustness and coverage-tests of non-sequential methods. We demonstrated that it allows to scale neural posterior estimation to complex and high-dimensional scientific problems.

## Acknowledgments and Disclosure of Funding

We thank Poornima Ramesh, Cornelius Schröder, Marcel Nonnenmacher, David Greenberg, and Jan-Matthis Lueckmann for discussions and feedback. We also thank the International Max Planck Research School for Intelligent Systems (IMPRS-IS) for supporting MD. This work was funded by the German Research Foundation (DFG; Germany's Excellence Strategy MLCoE – EXC number 2064/1 PN 390727645) and the German Federal Ministry of Education and Research (BMBF; Tübingen AI Center, FKZ: 01IS18039A).

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
