# 6 Appendix

## 6.1 Reproducibility statement

We used the configuration manager hydra to track the configuration and seeds of each run [Yadan, 2019]. All code to reproduce the results can be found at `https://github.com/mackelab/tsnpe_neurips`. We implemented TSNPE on top of the publicly accessible sbi toolbox [Tejero-Cantero et al., 2020]. All simulations and runs were performed on a high-performance computer. For each run, we used between 8 and 48 CPU cores.

## 6.2 Proof of convergence

Below, we prove that, for a given observation $\mathbf{x}_o$, the posterior distribution obtained with TSNPE $q_\phi(\boldsymbol{\theta}|\mathbf{x}_o)$ converges to the true posterior distribution $p(\boldsymbol{\theta}|\mathbf{x}_o)$ under the assumption that the HPR$_\epsilon$ of the approximate posterior at every TSNPE round covers the support (i.e., the HPR$_\epsilon$ for $\epsilon = 0$) of the true posterior given the observation $\mathbf{x}_o$. We call the support of the true posterior $\mathcal{M}$, i.e. $\mathcal{M} = \text{supp}(p(\boldsymbol{\theta}|\mathbf{x}_o))$. The proof proceeds in two steps: First, we derive the effective proposal distribution when pooling data from all rounds. Second, we show that, for such proposal distributions, the neural density estimator converges to the true posterior.

**Deriving the proposal distribution** We denote by $\tilde{p}^r(\boldsymbol{\theta})$ the proposal distribution from which $\boldsymbol{\theta}$ are drawn in round $r$. In the first round, we use the prior, i.e., $\tilde{p}^{r=1}(\boldsymbol{\theta}) = p(\boldsymbol{\theta})$. In later rounds, we sample from the prior but reject samples that lie outside of the HPR$_\epsilon$ of the approximate posterior given $\mathbf{x}_o$. The samples drawn in round $r$ are thus drawn from

$$\tilde{p}^r(\boldsymbol{\theta}) = U^r(\boldsymbol{\theta})p(\boldsymbol{\theta})/Z^r, \tag{1}$$

where $Z^r$ is the normalization constant and $U^r(\boldsymbol{\theta})$ is 1 on the HPR$_\epsilon$ of the approximate posterior at round $r$ and zero otherwise. Given our assumption that the HPR$_\epsilon$ of the approximate posterior covers the support of the true posterior $\mathcal{M}$, $U^r(\boldsymbol{\theta})$ is 1 on the support of the true posterior $\mathcal{M}$.

When pooling simulations from all rounds, after $R$ rounds, the parameters are sampled from a mixture of all proposal distributions:

$$\tilde{p}(\boldsymbol{\theta}) = \frac{1}{N}\sum_{r=1}^{R}\tilde{p}^r(\boldsymbol{\theta}) = p(\boldsymbol{\theta}) \cdot \left(\frac{1}{N}\sum_{r=1}^{R}U^r(\boldsymbol{\theta})/Z^r\right) = p(\boldsymbol{\theta}) \cdot f(\boldsymbol{\theta}). \tag{2}$$

In this equation, we assumed that all rounds contain equally many simulations, but the proof can easily be extended to rounds with different numbers of simulations by adding weights to the above sum.

As can be seen above, the distribution $\tilde{p}(\boldsymbol{\theta})$ is the prior times a function $f(\boldsymbol{\theta})$ which is made up of several steps and whose steps are defined by the HPR$_\epsilon$ of the approximate posterior of every round (illustration in Appendix Fig. 23). Finally, under the assumption that all $U^r(\boldsymbol{\theta})$ are 1 on the support of the true posterior $\mathcal{M}$, we have, for any $\boldsymbol{\theta} \in \mathcal{M}$

$$f(\boldsymbol{\theta}) = \frac{1}{N}\sum_{r=1}^{R}U^r(\boldsymbol{\theta})/Z_1^r = \frac{1}{N}\sum_{r=1}^{R}1/Z_1^r = \text{constant} = c,$$

i.e., $f(\boldsymbol{\theta})$ is not a function of $\boldsymbol{\theta}$. Thus, for any $\boldsymbol{\theta} \in \mathcal{M}$, we have

$$\tilde{p}(\boldsymbol{\theta}) = p(\boldsymbol{\theta}) \cdot c \propto p(\boldsymbol{\theta})$$

We emphasise that this proportionality holds only for $\boldsymbol{\theta} \in \mathcal{M}$, but not necessarily for $\boldsymbol{\theta}$ outside of the support of the true posterior.

**Training the neural density estimator** Next, we show that, for a proposal distribution of the form derived in the paragraph above, the approximate posterior $q_\phi(\boldsymbol{\theta}|\mathbf{x}_o)$ for an observation $\mathbf{x}_o$ converges to the true posterior distribution $p(\boldsymbol{\theta}|\mathbf{x}_o)$.

TSNPE minimizes the following loss function:

$$\mathcal{L} = -\frac{1}{N} \sum_i \log(q_\phi(\boldsymbol{\theta}_i|\mathbf{x}_i))$$

$$\xrightarrow{N\to\infty} -\mathbb{E}_{p(\boldsymbol{\theta},\mathbf{x})}[\log(q_\phi(\boldsymbol{\theta}|\mathbf{x}))]$$

$$= -\mathbb{E}_{\tilde{p}(\boldsymbol{\theta})p(\mathbf{x}|\boldsymbol{\theta})}[\log(q_\phi(\boldsymbol{\theta}|\mathbf{x}))].$$

Plugging in the proposal distribution defined above:

$$\mathcal{L} = -\iint f(\boldsymbol{\theta})p(\boldsymbol{\theta})p(\mathbf{x}|\boldsymbol{\theta})\log(q_\phi(\boldsymbol{\theta}|\mathbf{x}))\,d\boldsymbol{\theta}d\mathbf{x}$$

$$= \int p(\mathbf{x})\int -f(\boldsymbol{\theta})p(\boldsymbol{\theta}|\mathbf{x})\log(q_\phi(\boldsymbol{\theta}|\mathbf{x}))\,d\boldsymbol{\theta}d\mathbf{x}.$$

The term within the integral over $\boldsymbol{\theta}$ is proportional to the Kullback-Leibler-divergence between $f(\boldsymbol{\theta})p(\boldsymbol{\theta}|\mathbf{x})/Z$ (with $Z = \int f(\boldsymbol{\theta})p(\boldsymbol{\theta}|\mathbf{x})d\boldsymbol{\theta}$) and the approximate posterior $q_\phi(\boldsymbol{\theta}|\mathbf{x})$. Thus, $\mathcal{L}$ is minimized if and only if

$$q_\phi(\boldsymbol{\theta}|\mathbf{x}) \propto f(\boldsymbol{\theta})p(\boldsymbol{\theta}|\mathbf{x})$$

for all $\mathbf{x}$ within the the support of $p(\mathbf{x})$ [Papamakarios and Murray, 2016].

This means that, for arbitrary $\mathbf{x} \in \text{supp}(p(\mathbf{x}))$, $q_\phi(\boldsymbol{\theta}|\mathbf{x})$ will not converge to the true posterior $p(\boldsymbol{\theta}|\mathbf{x})$, but to $f(\boldsymbol{\theta})p(\boldsymbol{\theta}|\mathbf{x})/Z$. However, for the observed data $\mathbf{x} = \mathbf{x}_o$, we have:

$$q_\phi(\boldsymbol{\theta}|\mathbf{x}_o) \propto f(\boldsymbol{\theta})p(\boldsymbol{\theta}|\mathbf{x}_o) = \begin{cases} c \cdot p(\boldsymbol{\theta}|\mathbf{x}_o) & \text{if } \boldsymbol{\theta} \in \mathcal{M} \\ f(\boldsymbol{\theta}) \cdot 0 & \text{else} \end{cases}$$

The first case follows from the fact that $f(\boldsymbol{\theta})$ is constant on the support of the true posterior. The second case follows because the true posterior has zero probability density for $\boldsymbol{\theta}$ outside of its own support $\mathcal{M}$. Thus:

$$q_\phi(\boldsymbol{\theta}|\mathbf{x}_o) \propto p(\boldsymbol{\theta}|\mathbf{x}_o).$$

Since $q_\phi(\boldsymbol{\theta}|\mathbf{x}_o)$ is a (conditional) normalizing flow, it is normalized and, thus:

$$q_\phi(\boldsymbol{\theta}|\mathbf{x}_o) = p(\boldsymbol{\theta}|\mathbf{x}_o).$$

### 6.3 Simulation-based coverage calibration (SBCC)

The algorithm for computing the coverage (SBCC) is shown in Alg. 2.

---

**Algorithm 2:** Simulation-based coverage calibration (SBCC)

---

**Inputs:** proposal $\tilde{p}(\boldsymbol{\theta})$, approximate posterior $q_\phi$ trained on $\boldsymbol{\theta} \sim \tilde{p}(\boldsymbol{\theta})$, number of simulations $M$, number of posterior samples per simulation $P$. Database of empirical coverages $\mathcal{E}$, initialized as the empty set.

**Outputs:** Coverage for different confidence levels $1 - \alpha$.

**for** $i \in [1, ..., M]$ **do**

    $\boldsymbol{\theta}_i^* \sim \tilde{p}(\boldsymbol{\theta})$

    simulate $\mathbf{x}_i^* \sim p(\mathbf{x}|\boldsymbol{\theta}_i^*)$

    $l_i^* = \log(q_\phi(\boldsymbol{\theta}_i^*|\mathbf{x}_i^*))$ ;           // compute log-prob of ground-truth

    $c = 0$

    **for** $j \in [1, ..., P]$ **do**

        $\boldsymbol{\theta}_j \sim q_\phi(\boldsymbol{\theta}|\mathbf{x}_i^*)$

        $l_j = \log(q_\phi(\boldsymbol{\theta}_j|\mathbf{x}_i^*))$ ;       // compute log-probs of posterior samples

        **if** $l_j > l_i^*$ **then**

            $c = c + 1$

        $e = c/P$ ;  // fraction of posterior samples whose log-prob is larger than ground-truth log-prob

        add $e$ to $\mathcal{E}$

Plot CDF of $\mathcal{E}$

---

---

**Algorithm 3:** Obtaining the HPR$_\epsilon$ of the approximate posterior and sampling the truncated proposal with rejection sampling

---

**Inputs:** Prior $p(\boldsymbol{\theta})$, Approximate posterior $q_\phi$, observation $\mathbf{x}_o$, number of posterior samples $M$, $\epsilon$ that defines the HPR, number of desired samples from $\tilde{p}(\theta)$ $N$. Database of log-probabilities $\mathcal{P}$, initialized as an empty set.

**Outputs:** Samples $\mathcal{S}$ from truncated proposal $\tilde{p}(\boldsymbol{\theta})$.

**for** $i \in [1, ..., M]$ **do**
 | $\boldsymbol{\theta}_i \sim q_\phi(\boldsymbol{\theta}|\mathbf{x}_o)$
 | $l_i = \log(q_\phi(\boldsymbol{\theta}_i|\mathbf{x}_o)$
 | add $l_i$ to $\mathcal{P}$
$\boldsymbol{\kappa} = \text{quantile}_\epsilon(\mathcal{P})$ ;          `// threshold for HPR`$_\epsilon$

`// sample truncated proposal with rejection sampling`
**while** $s < N$ **do**
 | $\boldsymbol{\theta} \sim p(\boldsymbol{\theta})$
 | $l = \log(q_\phi(\boldsymbol{\theta}|\mathbf{x}_o))$
 | **if** $l > \boldsymbol{\kappa}$ **then**
  | add $\boldsymbol{\theta}$ to $\mathcal{S}$
  | s += 1

---

### 6.4 Algorithm description for sampling from the HPR of the approximate posterior

The algorithm for sampling from the HPR$_\epsilon$ of the approximate posterior is shown in Alg. 3.

### 6.5 Alleviating issues of APT

We compared TSNPE to Automatic Posterior Transformation (APT) [Greenberg et al., 2019]. In this section, we briefly review how APT works, and why 'leakage' occurs, and how we attempted to improve it.

**APT review** Broadly, APT exists in two versions. Its first version can be applied only if the density estimator $q_\phi(\boldsymbol{\theta}|\mathbf{x})$ is a mixture of Gaussians, the proposal $\tilde{p}(\boldsymbol{\theta})$ is a mixture of Gaussians, and the prior $p(\boldsymbol{\theta})$ is either uniform or Gaussian. We did not compare TSNPE to APT in this form because we wanted to use more expressive density estimators for $q_\phi(\boldsymbol{\theta}|\mathbf{x})$. The second version of APT, known as atomic APT, allows to use any density estimator $q_\phi(\boldsymbol{\theta}|\mathbf{x})$, any proposal $\tilde{p}(\boldsymbol{\theta})$ and any explicit prior $p(\boldsymbol{\theta})$. One must be able to evaluate the density estimator and the prior, but the proposal can be implicit (i.e., without closed form density). Atomic APT minimizes the loss:

$$\mathcal{L}_\phi = -\frac{1}{N} \sum_{i=1}^{N} \log \frac{q_\phi(\boldsymbol{\theta}_i|\mathbf{x}_i)/p(\boldsymbol{\theta}_i)}{q_\phi(\boldsymbol{\theta}_i|\mathbf{x}_i)/p(\boldsymbol{\theta}_i) + \sum_{j=1...A-1} q_\phi(\boldsymbol{\theta}_j|\mathbf{x}_i)/p(\boldsymbol{\theta}_j)}$$

with number of atoms $A$. In this loss, $\boldsymbol{\theta}_j$ and $\boldsymbol{\theta}_i$ can be sampled from *any* proposal distribution. $\mathbf{x}_i$ is sampled from $p(\mathbf{x}|\boldsymbol{\theta}_i)$.

**Leakage issue** Notice that the above loss is the same for $q_\phi(\boldsymbol{\theta}|\mathbf{x})$ and for $c \cdot q_\phi(\boldsymbol{\theta}|\mathbf{x})$. In other words, the approximate posterior has to be correct only up to a proportionality constant [Greenberg et al., 2019, Durkan et al., 2020]. Thus, an approximate posterior $q_\phi(\boldsymbol{\theta}|\mathbf{x})$ which is $c \cdot p(\boldsymbol{\theta}|\mathbf{x})$ will have a minimal loss. Since the dataset on which $q_\phi(\boldsymbol{\theta}|\mathbf{x})$ is trained contains no $\boldsymbol{\theta}$ that lies outside of the prior bounds, the approximate posterior can be anything outside of the prior bounds without affecting the value of the loss function. This is what is called 'leakage', and which has been pointed out as a potential problem both in the original APT paper Greenberg et al. [2019] and work studying the relationship of APT with contrastive learning approaches Durkan et al. [2020]: While the approximate posterior might be proportional to the true posterior within the bounds of the prior $p(\boldsymbol{\theta})$, it can put significant mass outside of the prior bounds (because the loss does not penalize this behaviour).

**Transforming the parameter space** We tried to fix the leakage issue by appending a transformation such that the density estimator has constrained support. This requires that the bounds of the parameter

space are known and that such a transformation can be implemented. If this fix can be applied, the leakage will be zero by definition. In all our experiments, such a transformation substantially helped with the leakage problem.

**Leakage into regions of no training data**    However, even when possible, such transformation does not completely prevent 'leakage': When inspecting the loss of APT, we see that the approximate posterior can put mass into any region of the parameter space in which no parameter sets $\boldsymbol{\theta}$ in the training data lie. For example, if the prior distribution is standard Gaussian and one trains APT on 100 parameter sets sampled from the prior, the training dataset will unlikely contain values that are smaller than -3 or larger than 3. Therefore, APT can 'leak' into the regions below -3 or above 3 and still have an optimal loss. While, in the limit of infinite data, APT would correct its 'leakage' as soon as parameters from these regions are used as training data, given a finite number of simulations, the approximate posterior can 'leak' into new regions which have not been explored (yet). The (potential) problem of leakage does not affect all applications equally: High-dimensional parameter spaces suffer more from this behavior than low-dimensional ones, as there are many regions into which the mass of the approximate posterior can 'leak'. This behaviour is illustrated in Appendix Fig. 17.

**Leakage because of invalid data**    Another way in which leakage can occur is if the simulator produces invalid data (e.g. NaN or infinity). Often, such invalid simulations are discarded from the training dataset [Lueckmann et al., 2017] and the approximate posterior is trained only on samples that produce valid outputs. For example, assume that a small region of the prior always produces invalid simulations. In this case, the approximate posterior will never be trained on simulations from this parameter region and APT can 'leak' into this region. Thus, the approximate posterior might contain significant mass in parameter regions that produce invalid simulations.

**Explicit recommendations for running APT**    We will now give explicit recommendations for running APT. These modification greatly improved the performance of APT in our experiments, but were not able to avoid the failure of the algorithm on challenging real-world problems such as the pyloric network (Fig. 5).

1. For priors with bounded supports: If possible, transform the parameter space into unbounded space.

2. Do not discard invalid simulations. Instead, replace invalid entries (such as NaN) with a substantially different value than the observed data and train on all available simulations.

3. Check if the posterior contains a lot of mass in regions with very low prior probability. If this is the case, it can hint at a failure of APT through leakage into regions of no training data.

4. If you transformed the parameter space, check if many posterior samples lie very close to the bounds. Again, this can hint at a failure of APT through leakage into regions of no training data.

## 6.6    Relation between simulation-based calibration and our diagnostic

As discussed above, our method is closely related to simulation-based calibration (SBC) [Cook et al., 2006, Talts et al., 2018]. Briefly, SBC samples $\boldsymbol{\theta}^*$ from the prior, samples the likelihood $\mathbf{x}^* \sim p(\mathbf{x}|\boldsymbol{\theta}^*)$ and then draws samples from the posterior $\boldsymbol{\theta}_i \sim p(\boldsymbol{\theta}|\mathbf{x}^*)$ (e.g., with MCMC). It then projects the (potentially high-dimensional) parameters $\boldsymbol{\theta}_i$ into a one-dimensional space $T(\boldsymbol{\theta}_i)$. Often, this projection is the 1D-marginal distribution of parameters [Carpenter et al., 2017]. It then ranks $T(\boldsymbol{\theta}^*)$ under all posterior samples $T(\boldsymbol{\theta}_{1...N})$. Repeated across several prior samples, the distribution of ranks should be uniform. For high-dimensional parameter spaces, the marginal distribution of each parameter is checked independently.

Notably, other projections into a one-dimensional space are possible. Below, we explain that our method is identical to running SBC with projection $T : \boldsymbol{\theta} \to q_\phi(\boldsymbol{\theta}|\mathbf{x})$ and with posterior samples exactly following $q_\phi(\boldsymbol{\theta}|\mathbf{x})$.

In SBC and in our diagnostic method, samples are drawn from the prior $\boldsymbol{\theta}^* \sim p(\boldsymbol{\theta})$ and simulated $\mathbf{x}^* \sim p(\mathbf{x}|\boldsymbol{\theta})$. In our diagnostic method, as well as in SBC with projection being the log-probability

of the approximate posterior, one then samples the posterior to obtain $\boldsymbol{\theta}_i$ and evaluates the log-probability of all samples, i.e., $l_i = T(\boldsymbol{\theta}_i) = \log(q(\boldsymbol{\theta}_i|\mathbf{x}^*)$ as well as of the initial parameter set $l^* = \log(q(\boldsymbol{\theta}^*|\mathbf{x}^*)$. SBC then ranks $l^*$ under all $l_i$, which is equivalent to computing its quantile (as in our method). In our diagnostic tool, one then evaluates whether this quantile is above or below several confidence levels (evaluation on a 1D evenly spaced grid, same as rank binning in SBC). This generates a step-function with the step occuring at the quantile of $l^*$. This step function is the cumulative distribution function of a dirac at the quantile of $l^*$. Therefore, repeated across several $\boldsymbol{\theta}^*$, our coverage plots (e.g., Fig 3c) correspond to the cumulative distribution function of the histograms generated by SBC (with projection $T : \boldsymbol{\theta} \rightarrow \log(q_\phi(\boldsymbol{\theta}|\mathbf{x}))$ and posterior samples exactly following $q_\phi(\boldsymbol{\theta}|\mathbf{x})$).

## 6.7   SBCC in a multi-round setting

We run our diagnostic tool after every round of training. If one trains only on simulations that were run in the most recent round, SBCC can be run as in the first round. However, if one wishes to train on simulations from all rounds, then the deep neural density estimator converges to:

$$q_\phi(\boldsymbol{\theta}|\mathbf{x}) \propto f(\boldsymbol{\theta})p(\boldsymbol{\theta}|\mathbf{x}).$$

Proof in Sec. 6.2. As is described in Sec. 6.2, this means that $q_\phi(\boldsymbol{\theta}|\mathbf{x})$ will not converge to the true posterior for arbitrary $\mathbf{x}$, but only for the observation $\mathbf{x}_o$.

This poses a problem for SBCC: As described in Alg. 2, SBCC measures whether the coverage is correct (on average) for many $\mathbf{x}$ generated by the proposal distribution. Since the loss employed by TSNPE only ensures convergence for $\mathbf{x}_o$, it will by construction not provide correct results for other $\mathbf{x}$.

This issue can be solved in two ways:

1. When running SBCC, instead of drawing samples from the proposal prior of the most recent round $\tilde{p}^r(\boldsymbol{\theta})$, one can draw samples $\boldsymbol{\theta}^*$ from $\tilde{p}(\boldsymbol{\theta})$, i.e., the distribution that emerges from pooling data from all rounds (notation as in Sec. 6.2 and Alg. 2). This is the method described in Appendix Alg. 2.

2. One can truncate the approximate posteriors $q_\phi(\boldsymbol{\theta}|\mathbf{x}^*)$ while running SBCC (see Alg. 2 for notation). With this strategy, when running SBCC in round $r$, we draw parameters from $\boldsymbol{\theta}^* \sim \tilde{p}^r(\boldsymbol{\theta})$ (the truncated proposal from round $r$), simulate them $\mathbf{x}^* \sim p(\mathbf{x}|\boldsymbol{\theta}^*)$, sample from the posterior $\boldsymbol{\theta}_i \sim q_\phi(\boldsymbol{\theta}|\mathbf{x}^*)$, reject samples that lie outside of $\text{HPR}_\epsilon(q_\phi(\boldsymbol{\theta}_i|\mathbf{x}_o))$, and then continue as described in Alg. 2. Strategy 2 only ensures that the posterior regions which lie within $\text{HPR}_\epsilon(q_\phi(\boldsymbol{\theta}|\mathbf{x}_o))$ are well-calibrated. It does not ensure that the full posterior ($q_\phi(\boldsymbol{\theta}|\mathbf{x}^*)$ with $\mathbf{x}^* \sim p(\mathbf{x}^*|\boldsymbol{\theta}^*)$ and $\boldsymbol{\theta}^* \sim \tilde{p}^r(\boldsymbol{\theta})$) is well-calibrated.

## 6.8   Toy model

The toy model shown in Fig. 1 is given by a uniform prior within $[-2, -1]$ and $[1, 2]$. The simulator is $x \sim \theta^2 + \epsilon$, where $\epsilon$ is a Gaussian distribution with mean zero and standard deviation $0.2$. We ran APT [Greenberg et al., 2019] and TSNPE for 5 rounds with 500 simulations per round. For APT, all hyperparameters are the default values from the sbi package [Tejero-Cantero et al., 2020], but we used a neural spline flow (NSF) for both APT and TSNPE [Durkan et al., 2019].

## 6.9   Benchmark tasks

Below, we briefly describe the benchmark tasks. For details, please see Lueckmann et al. [2021].

**Gaussian linear:** 10 parameters which are the mean of a Gaussian model. The prior is Gaussian, resulting in a Gaussian posterior.

**Bernoulli GLM:** Generalized linear model with Bernoulli observations. Inference is performed on 10-dimensional sufficient summary statistics of the originally 100 dimensional raw data. The resulting posterior is 10-dimensional, unimodal, and concave.

**Lotka Volterra:** A traditional model in ecology [Wangersky, 1978], which describes a predator-prey interaction between species, illustrating a task with complex likelihood and unimodal posterior.

**SLCP:** A task introduced by Papamakarios et al. [2019] with a simple likelihood and complex posterior. The prior is uniform, the likelihood has Gaussian noise but is non-linearly related to the parameters, resulting in a posterior with four symmetrical modes.

**Two moons:** This model has two parameters with a uniform prior. The simulator is non-linear, generating a posterior with both local and global (bimodal) structure [Greenberg et al., 2019].

**SIR:** Epidemiological model with two parameters and ten summary statistics [Kermack and McKendrick, 1927].

### 6.10 Errors due to truncation

As described in Appendix Sec. 6.2, the approximate posterior converges to the true posterior if the truncated proposal covers the support of the true posterior. The truncated support is defined as the high probability region that contains 1-$\epsilon$ of mass of the approximate posterior (HPR$_\epsilon$). For $\epsilon > 0$, the HPR$_\epsilon$ will likely not be a superset of the support of the true posterior and hence, there will be errors in posterior approximation. In this section, we discuss the effect of these errors on inference accuracy.

When the value of $\epsilon$ is chosen too large, the tails of the approximate posterior are excluded from the truncated proposal. In the following training round, the approximate posterior converges to a distribution that is correct, up to proportionality, within the HPR$_\epsilon$ of the previous approximate posterior, but that underestimates the tails of the posterior distribution. We demonstrate this behavior in Appendix Fig. 11 for a linear Gaussian simulator, uniform prior, 50k simulations per round, and a neural spline flow with 20 bins as neural density estimator. After round 1, the approximate posterior closely matches the true posterior (Fig. 11a). When using a large $\epsilon$, e.g., $\epsilon = 0.1$, the proposal for the second round is narrower than the true posterior and, thus, the proposal obtained by pooling data from both rounds is not constant on the support of the true posterior (Fig. 11b, blue). This leads to the approximate posterior underestimating the tails of the true posterior (Fig. 11b, purple). When using a smaller $\epsilon$, e.g., $\epsilon = 0.01$, the errors induced by truncation become small and inference errors are mostly due to finite data and imperfect convergence of the neural network (Fig. 11c). We note that, throughout our study, we evaluated $\epsilon = \{10^{-3}, 10^{-4}, 10^{-5}\}$, i.e., values that are at least one order of magnitude smaller than $\epsilon = 0.01$.

Overall, this analysis demonstrates that the truncation performed by TSNPE can negatively impact inference quality in the tails of the posterior distribution. We, thus, do not recommend TSNPE in scenarios in which users are particularly interested in the tails of the posterior. In all our benchmark tasks, however, we did not find that the truncation negatively impacted inference quality as measured by C2ST accuracy (Fig. 4, Appendix Fig. 8, Fig. 8). This indicates that, for many (real-world) tasks, the errors due to truncation are outweighed by errors due to finite simulation budgets or imperfect convergence of neural network training.

### 6.11 Computational cost of rejection sampling and SIR

In this section, we quantify the computational costs of rejection sampling and sampling-importance resampling (SIR). The computational cost of both sampling methods comprises the computational cost of sampling and evaluating the approximate posterior. On an AMD Ryzen Threadripper 1920X 12-Core Processor, drawing (or evaluating) 100k samples takes approximately 10 seconds. On a GeForce RTX 2080 GPU, drawing (or evaluating) 100k samples takes approximately 0.17 seconds. Thus, in SIR (with an oversampling factor $K = 1024$), one can draw (or evaluate) 100 samples from the truncated proposal in 0.17 seconds on a GPU (versus 10 seconds on a CPU). For most real-world simulators, this constitutes a small fraction of the compute time required to simulate the model: e.g., for the multicompartment model, a single simulation takes approximately 30 seconds and, thus, SIR sampling from the truncated support takes up only 0.012% of the total compute time with a GPU. For rejection sampling, the time required to draw samples from the truncated support depends on the rejection rate. However, as long as the acceptance rate is above 0.001%, the cost of rejection sampling is still small compared to the cost of running the simulator.

### 6.12 Accuracy of SIR

Here, we investigate the (potential) error induced by using sampling-importance resampling (SIR). SIR is an approximate sampling technique and does not produce exact samples for finite $K$. This

raises the question of how strongly the errors induced by SIR influence the results of TSNPE. In order to investigate this, we performed three analyses: 1) We ran all benchmarking tasks with SIR and $K = 1024$ and compared the results to rejection sampling. Across all benchmark tasks, the performance of TSNPE with SIR matches the performance of TSNPE with rejection sampling (Appendix Fig. 10). 2) In a simple 1D toy model, we investigated how closely the samples produced by SIR match the samples produced by rejection sampling. As can be seen in Appendix Fig. 12, the distribution of SIR samples is quite different from rejection sampling for $K = 16$ and $K = 64$. However, for $K = 1024$, the distribution of samples from SIR very closely matches the distribution of rejection samples. 3) Finally, we investigated the performance of SIR by inspecting the effective sample size (ESS), which we computed as

$$\text{ESS} = 1/\sum_i^K w_i^2$$

with $w_i$ being the normalized importance weights [Kong, 1992, Djuric et al., 2003, Martino et al., 2017]. For $K = 1024$, across all benchmark tasks, the ESS was on average 25.154 and was never below 2.547, i.e. it was always significantly higher than 1 (the number of resampled samples). All of these results indicate that SIR is expected to be a useful and robust sampling method for TSNPE.

### 6.13 Pyloric network model

For the pyloric network model, we used the same prior, simulator, and summary statistics as previous work [Gonçalves et al., 2020, Deistler et al., 2021, Glöckler et al., 2022]. The model has a total of 31 parameters and 18 summary statistics. We replaced invalid summary statistics with a value that is 2 standard deviations (of prior predictives) below the observation. The experimental data [Haddad and Marder, 2021] is also the same as used in these previous works.

### 6.14 Multicompartment model of a single neuron

We performed Bayesian inference in a complex model of single-neuron dynamics. The model is the same as used in Van Geit et al. [2016]. We added observation noise with standard deviations taken from previously published measurements [Hay et al., 2011]. The prior is a uniform distribution within the same bounds as previously used [Van Geit et al., 2016]. The parameters are shown in Table 1. The summary statistics are also the same as in Van Geit et al. [2016].

We replaced NaN values by the minimal value among prior samples minus two standard deviations of prior samples. Several summary statistics had heavy tailed distributions, which led to very high standard deviations. For these summary statistics, we picked the replacement value by hand. The final values are shown in Table 2.

### 6.15 Choices of hyperparameters

For the results on the benchmark tasks, we picked the same hyperparameters for TSNPE as those that were used in Lueckmann et al. [2021] for APT (called SNPE in Lueckmann et al. [2021]).

For both neuroscience tasks, we used a neural spline slow (NSF) as density estimator [Durkan et al., 2019]. The hyperparameters of the NSF are the defaults from the 'sbi' package [Tejero-Cantero et al., 2020]. On both of these tasks, we used $\epsilon = 10^{-3}$. All other hyperparameters are the defaults from the 'sbi' package with one exception: We used a batchsize of $500$ (instead of the default value 50).

For the pyloric network task, we ran APT with 2 atoms. We reduced the number of atoms from the default value in the 'sbi' package (10 atoms) because a larger number of atoms increased training time. When using 10 atoms, the training time exceeded the simulation time of the model on the pyloric network task. With 2 atoms, the training time of APT was comparable to the training time of TSNPE (albeit still a bit higher). We implemented the transformation of the parameter space with the pytorch method 'biject_to()' [Paszke et al., 2019]. For TSNPE, we initially sampled from the truncated proposal with rejection sampling. After the eighth round of training, the rejection rate became exceedingly high and we switched sampling importance resampling (SIR, with $K = 1024$ see Sec. 3.2).

For the multicompartment model of single-neuron dynamics, we initially sampled parameters from the truncated proposal with rejection sampling and switched to SIR after the third round. For this

| Index | Parameter | Ground truth |
|---|---|---|
| $\theta_1$ | gnats2_tbar_nats2_t_apical | 0.026145 |
| $\theta_2$ | gskv3_1bar_skv3_1_apical | 0.004226 |
| $\theta_3$ | gimbar_im_apical | 0.000143 |
| $\theta_4$ | gnata_tbar_nata_t_axonal | 3.137968 |
| $\theta_5$ | gk_tstbar_k_tst_axonal | 0.089259 |
| $\theta_6$ | gamma_cadynamics_e2_axonal | 0.00291 |
| $\theta_7$ | gnap_et2bar_nap_et2_axonal | 0.006827 |
| $\theta_8$ | gsk_e2bar_sk_e2_axonal | 0.007104 |
| $\theta_9$ | gca_hvabar_ca_hva_axonal | 0.00099 |
| $\theta_{10}$ | gk_pstbar_k_pst_axonal | 0.973538 |
| $\theta_{11}$ | gskv3_1bar_skv3_1_axonal | 1.021945 |
| $\theta_{12}$ | decay_cadynamics_e2_axonal | 287.19873 |
| $\theta_{13}$ | gca_lvastbar_ca_lvast_axonal | 0.008752 |
| $\theta_{14}$ | gamma_cadynamics_e2_somatic | 0.000609 |
| $\theta_{15}$ | gskv3_1bar_skv3_1_somatic | 0.303472 |
| $\theta_{16}$ | gsk_e2bar_sk_e2_somatic | 0.008407 |
| $\theta_{17}$ | gca_hvabar_ca_hva_somatic | 0.000994 |
| $\theta_{18}$ | gnats2_tbar_nats2_t_somatic | 0.983955 |
| $\theta_{19}$ | decay_cadynamics_e2_somatic | 210.48529 |
| $\theta_{20}$ | gca_lvastbar_ca_lvast_somatic | 0.000333 |

Table 1: L5PC parameters.

task, in the first round of training, we used an ensemble of 10 neural networks. From the second round onward we used only a single neural network. For all other runs (toy example, benchmark, pyloric network), we did not use ensembles but always trained only a single network.

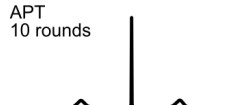

Figure 7: **APT performance after 10 rounds.** Same setup as in Fig. 1, but after running APT for 10 rounds. Leakage gets worse when additional rounds are run.

| Summary statistic | Observation | Replacement value |
|---|---|---|
| step1_soma_ahp_depth_abs | -62.1358 | -110.974 |
| step1_soma_ahp_depth_abs_slow | -62.2882 | -151.52 |
| step1_soma_ahp_slow_time | 0.140599 | -0.8473 |
| step1_soma_ap_height | 28.43591 | -33.959 |
| step1_soma_ap_width | 0.67857 | -2.132 |
| step1_soma_isi_cv | 0.03328 | -1.202 |
| step1_soma_adaptation_index2 | -0.0039499 | -0.3790 |
| step1_soma_doublet_isi | 67.00 | -1.699 |
| step1_soma_mean_frequency | 7.106 | -52.343 |
| step1_soma_time_to_first_spike | 33.3000 | -719.1 |
| step2_soma_ahp_depth_abs | -60.6933 | -110.974 |
| step2_soma_ahp_depth_abs_slow | -60.8186 | -151.066 |
| step2_soma_ahp_slow_time | 0.1496 | -0.8481 |
| step2_soma_ap_height | 26.5820 | -33.958 |
| step2_soma_ap_width | 0.67058 | -2.1747 |
| step2_soma_isi_cv | 0.03598 | -1.0649 |
| step2_soma_adaptation_index2 | -0.001467 | -0.47684 |
| step2_soma_doublet_isi | 44.600 | -1.6999 |
| step2_soma_mean_frequency | 8.8444 | -67.842 |
| step2_soma_time_to_first_spike | 23.000 | -719.1 |
| step3_soma_ahp_depth_abs | -56.759 | -110.744 |
| step3_soma_ahp_depth_abs_slow | -55.903 | -149.126 |
| step3_soma_ahp_slow_time | 0.2168 | -0.83586 |
| step3_soma_ap_height | 16.968 | -34.01 |
| step3_soma_ap_width | 0.5968 | -2.6932 |
| step3_soma_isi_cv | 0.09933 | -1.1164 |
| step3_soma_adaptation_index2 | 0.007206 | -0.5326 |
| step3_soma_doublet_isi | 21.100 | -1.699 |
| step3_soma_mean_frequency | 16.086 | -125.30 |
| step3_soma_time_to_first_spike | 10.600 | -719.1 |
| bap_dend1_ap_amplitude_from_voltagebase | 53.267 | -60.701 |
| bap_dend2_ap_amplitude_from_voltagebase | 30.592 | -31.779 |
| bap_soma_ap_height | 37.519 | -33.95 |
| bap_soma_ap_width | 0.800 | -1.583 |
| bap_soma_spikecount | 1.0 | -0.8855 |

Table 2: L5PC summary statistics.

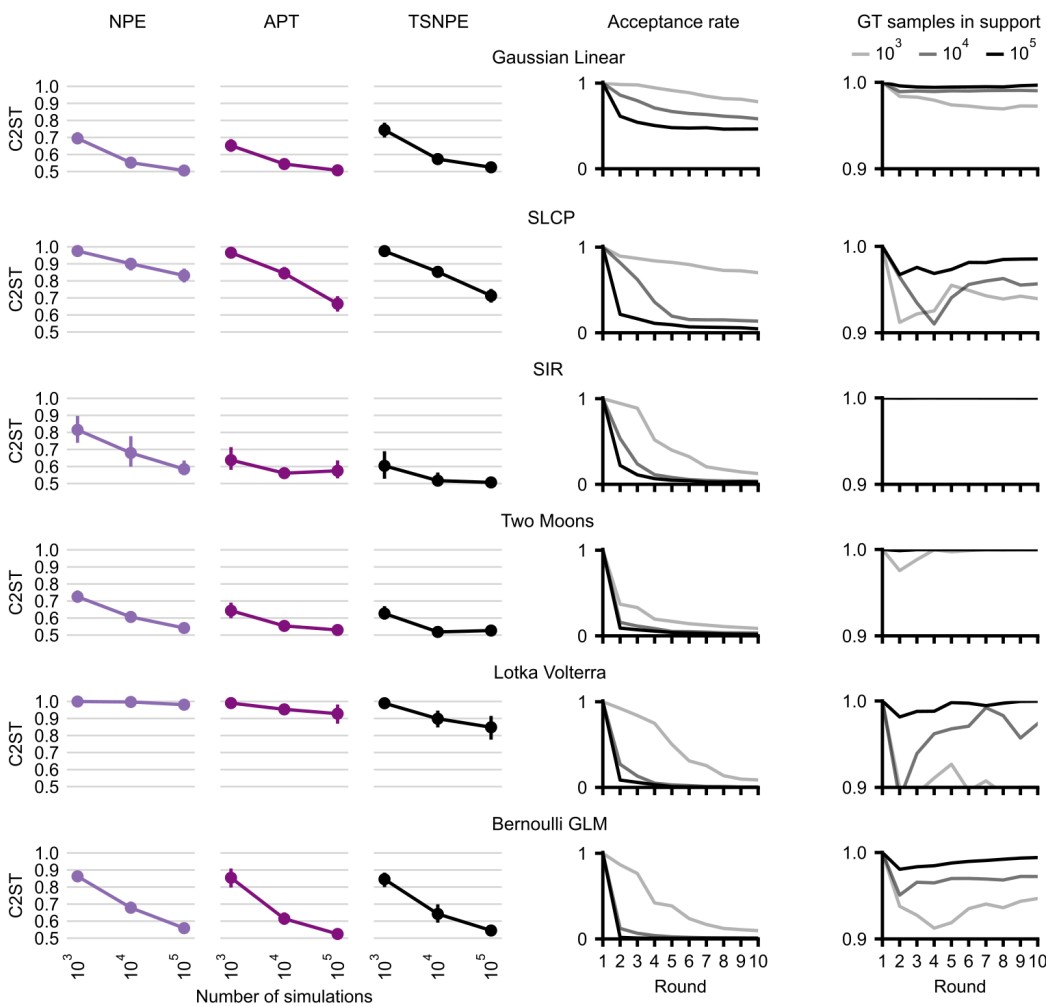

Figure 8: **Benchmark results for a less conservative threshold.** All hyperparameters are the same as in Fig. 4. The only difference is that we used $\epsilon = 10^{-3}$ (compared to $10^{-4}$ in Fig. 4).

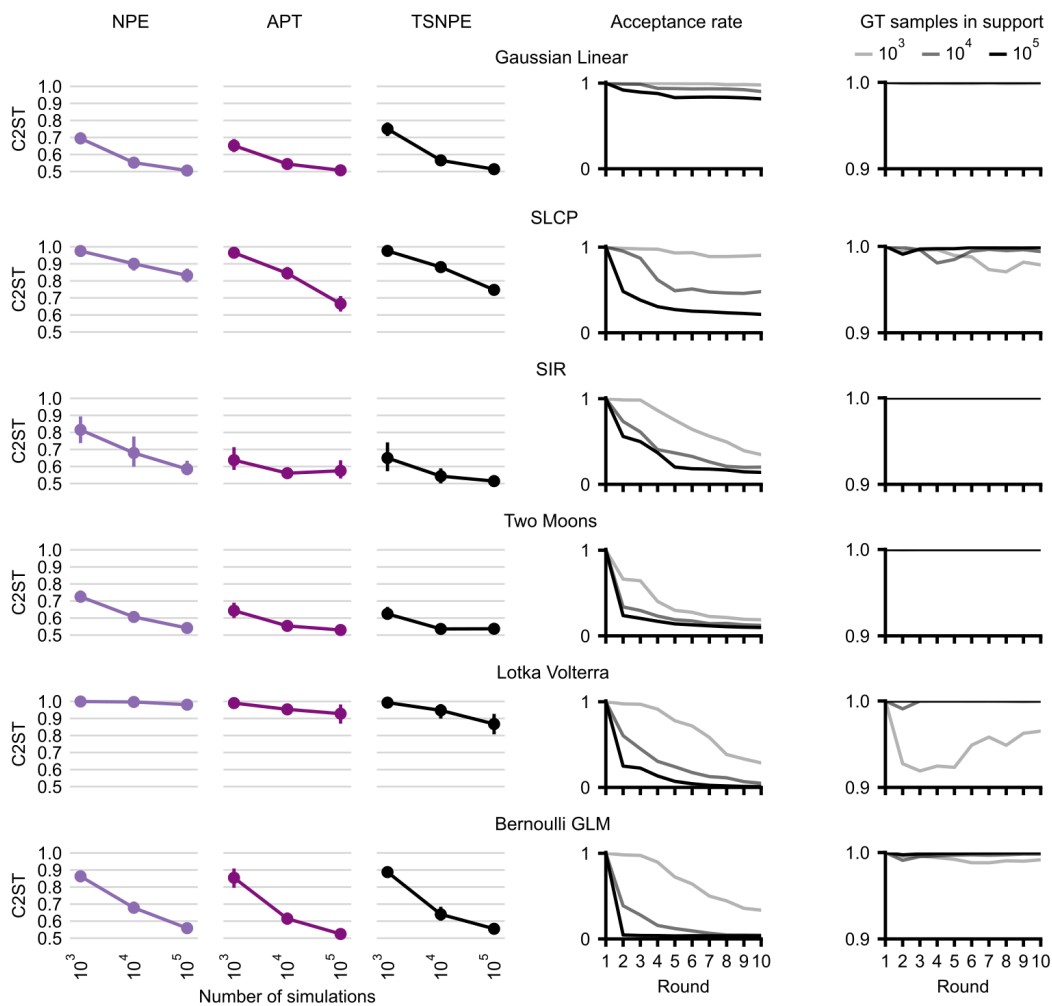

Figure 9: **Benchmark results for a more conservative threshold.** All hyperparameters are the same as in Fig. 4. The only difference is that we used $\epsilon = 10^{-5}$ (compared to $10^{-4}$ in Fig. 4).

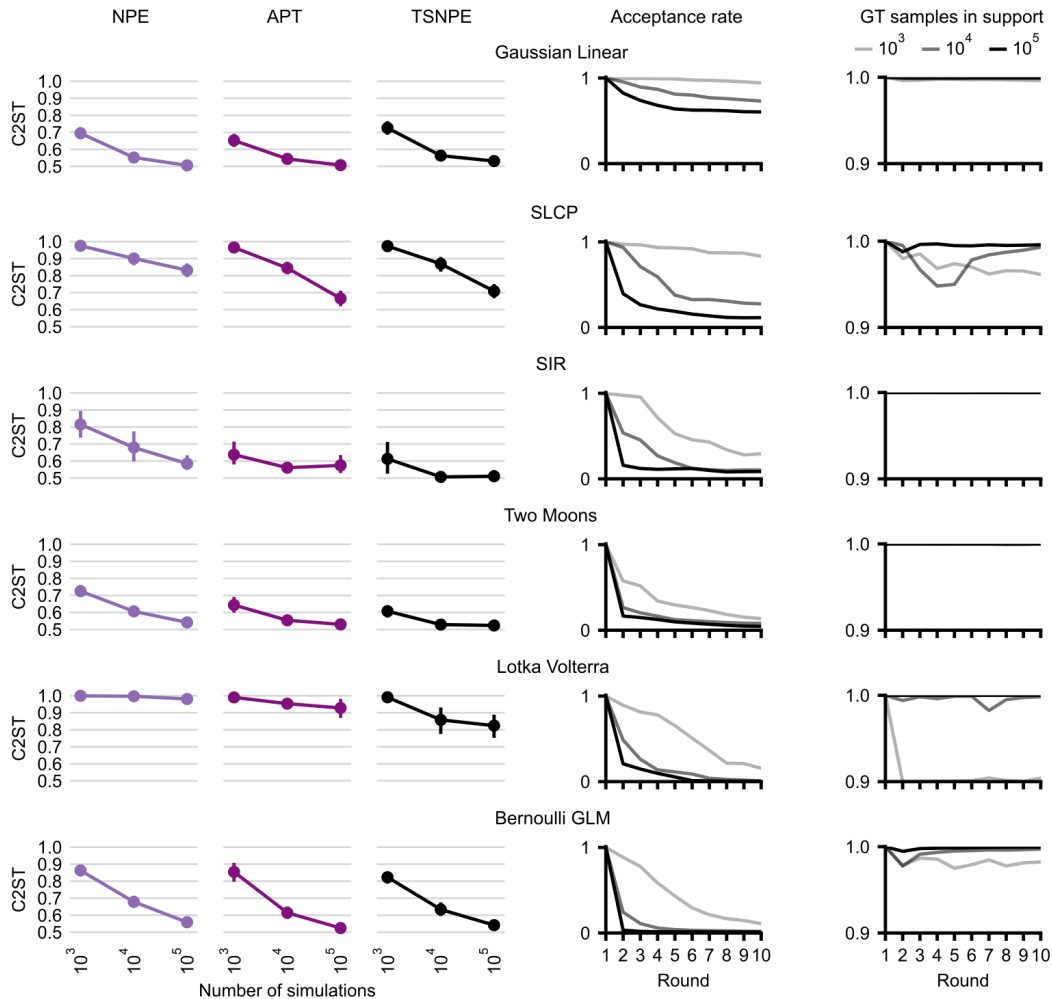

Figure 10: **Benchmark results for sampling importance resampling (SIR).** All hyperparameters are the same as in Fig. 4 ($\epsilon = 10^{-4}$) but we use SIR to sample from the truncated proposal (instead of rejection sampling in Fig. 4).

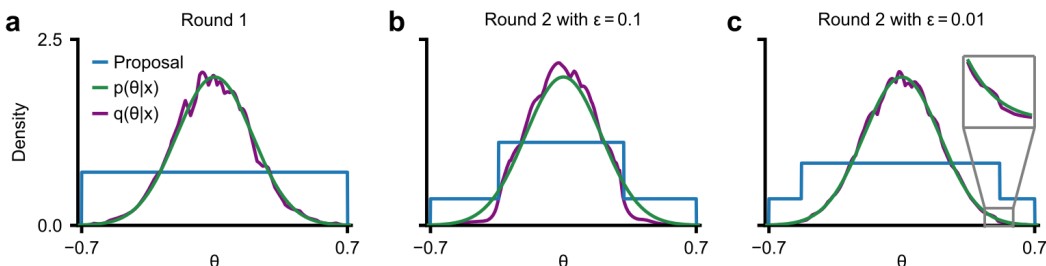

Figure 11: **Errors induced by truncation.** Inference in a linear Gaussian toy model with uniform prior. (a) The approximate posterior after round 1 closely matches the true posterior. (b) For a large truncation value $\epsilon = 0.1$, the approximate posterior after round 2 systematically underestimates the tails of the true posterior distribution. (c) For a smaller truncation value $\epsilon = 0.01$, the error induced by truncation is small.

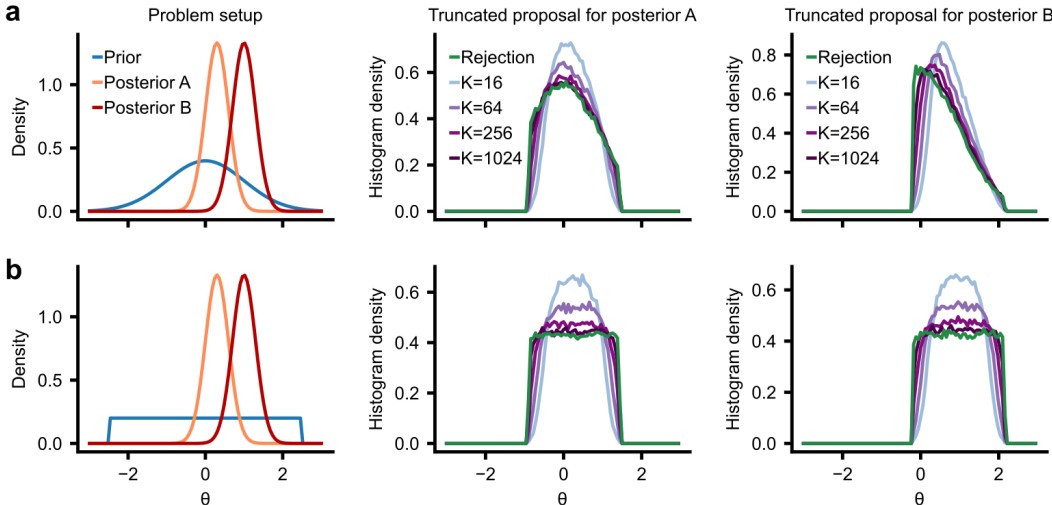

Figure 12: **Comparison of truncated proposal between rejection and importance sampling.** (a) Left: Gaussian prior as well as two posteriors. Middle: Density of 100k samples from the truncated proposal for posterior A. Green is rejection sampling, purple shaded colors are SIR with different oversampling factors $K$. Right: Same as middle, but for posterior B. (b) Same as panel a, but for a uniform prior.

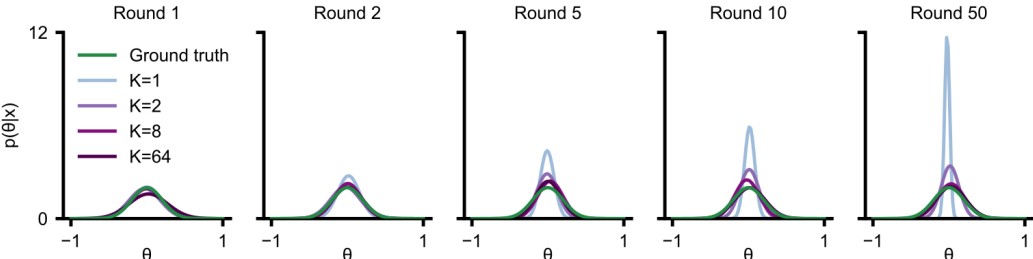

Figure 13: **Poor hyperparameter choices can lead SIR to diverge.** We applied TSNPE to a model with uniform prior (in [-1, 1]) and a linear Gaussian simulator. In each round, we ran 500 simulations, trained only on data from the most recent round, and used a Gaussian approximate posterior. We sampled from the truncated proposal with SIR (with different oversampling factors $K$). As more rounds are being run, the TSNPE approximate posterior can become too narrow for small $K$. Larger values of $K$ are robust across 50 rounds.

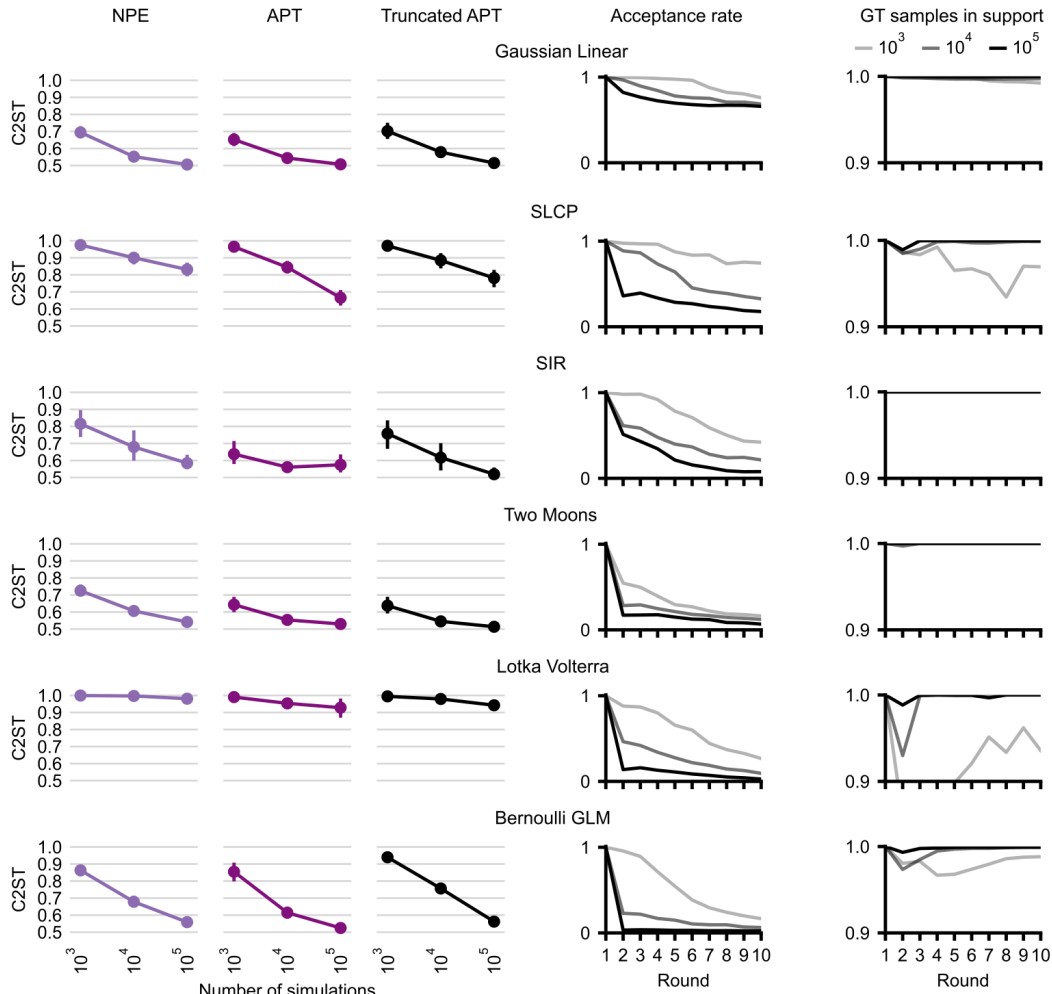

Figure 14: **Comparison between APT and truncated APT.** In order to investigate the effect of the truncated proposals on inference quality, we evaluated two versions of APT: In one scenario (middle column), we draw proposal samples from the previously estimated posterior, whereas in the second scenario (right column), we draw proposal samples from the truncated prior (with $\epsilon = 10^{-4}$). In both versions, we used the atomic loss function proposed in Greenberg et al. [2019]. Therefore, all differences stem from the different proposals. We also compare to standard NPE (trained with maximum-likelihood loss, left column). The two columns on the right are the same as in Fig. 4, evaluated for truncated APT.

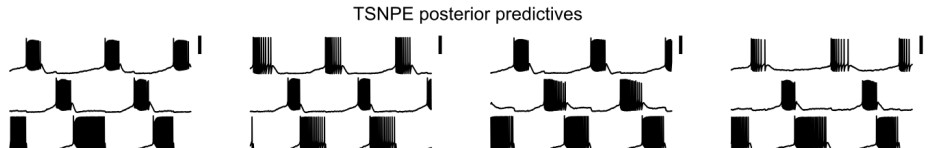

Figure 15: **Four posterior predictives of TSNPE applied to the pyloric network model.** The generated activity closely matches the summary statistics of the experimental data (Fig. 5a).

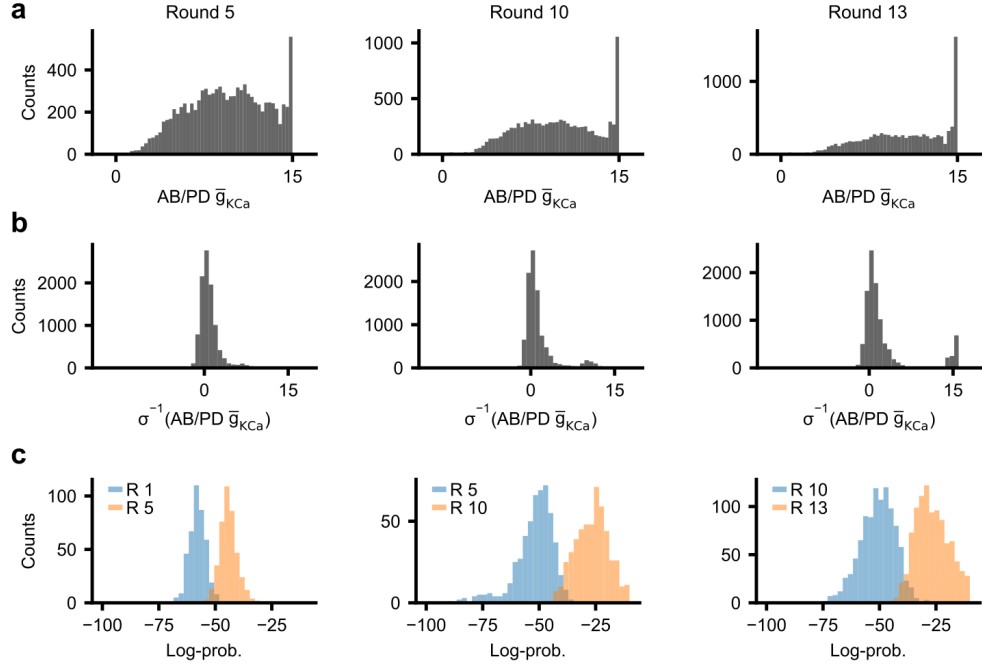

Figure 16: **Further explanation of issues when the parameter space is constrained.** (a) Marginal distribution of $\bar{g}_{KCa}$ of the AB/PD neuron when running APT for 5, 10, and 13 rounds. (b) Marginal distribution of this parameter, but when transformed into unconstrained space. This is the training data that the normalizing flow is 'effectively' seeing. (c) We evaluated those samples that were at the very right bounds of the marginal distribution under the current posterior (blue) as well as under the posterior from approximately 5 rounds before (orange). The samples have consistently lower log-probability under the previous posterior, which shows that the location of the leaking mass is moving.

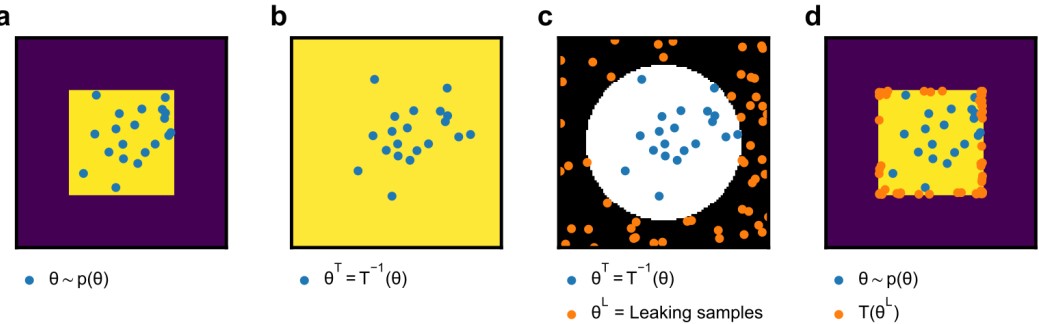

Figure 17: **Illustration of 'leakage' when prior is transformed to unconstrained space.** Note that these results are not based on an actual run of APT, but are purely for illustrative purposes. (a) The prior is a two-dimensional distribution on a constrained space (yellow region). Samples from the prior are in blue. (b) When the prior is transformed with an inverse sigmoid, its support becomes unconstrained. (c) APT will fit the posterior within the region of the training data (white region, blue samples). Outside of the training data, the trained density estimator can put arbitrary mass without affecting the loss (black region). Samples from this region are in orange. (d) When these 'leaking' samples (obtained with APT) are transformed back into constrained space, the 'leaking' mass ends up on the bounds of the prior.

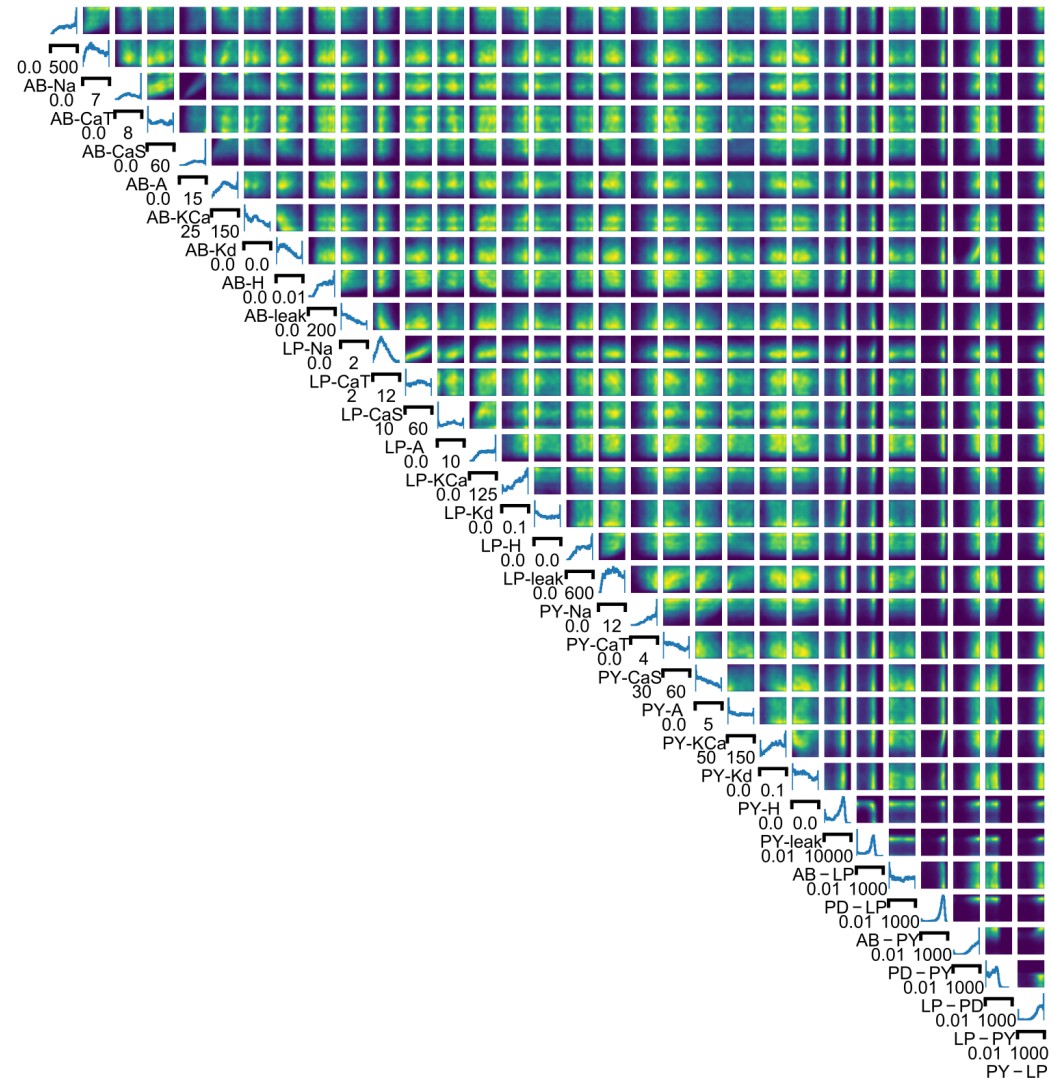

Figure 18: **Posterior distribution inferred by APT for the pyloric network model.**

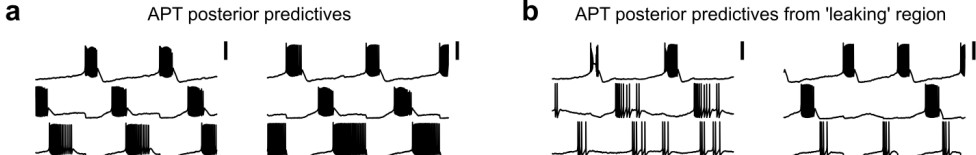

Figure 19: **Failure of APT when transformed to constrained space.** (a) Activity generated by two parameter sets sampled from the posterior distribution obtained with APT. We ensured that the shown parameter sets are not sampled from the very bounds of posterior distribution (i.e., that they were not sampled from the peak shown in Fig. 5c). (b) Activity generated by two parameter sets sampled from the posterior distribution obtained with APT. We specifically selected parameter sets whose value of $\overline{g}_{KCa}$ in the AB-PD neuron was above 14.999 (i.e., those samples which are in the peak shown in Fig. 5c).

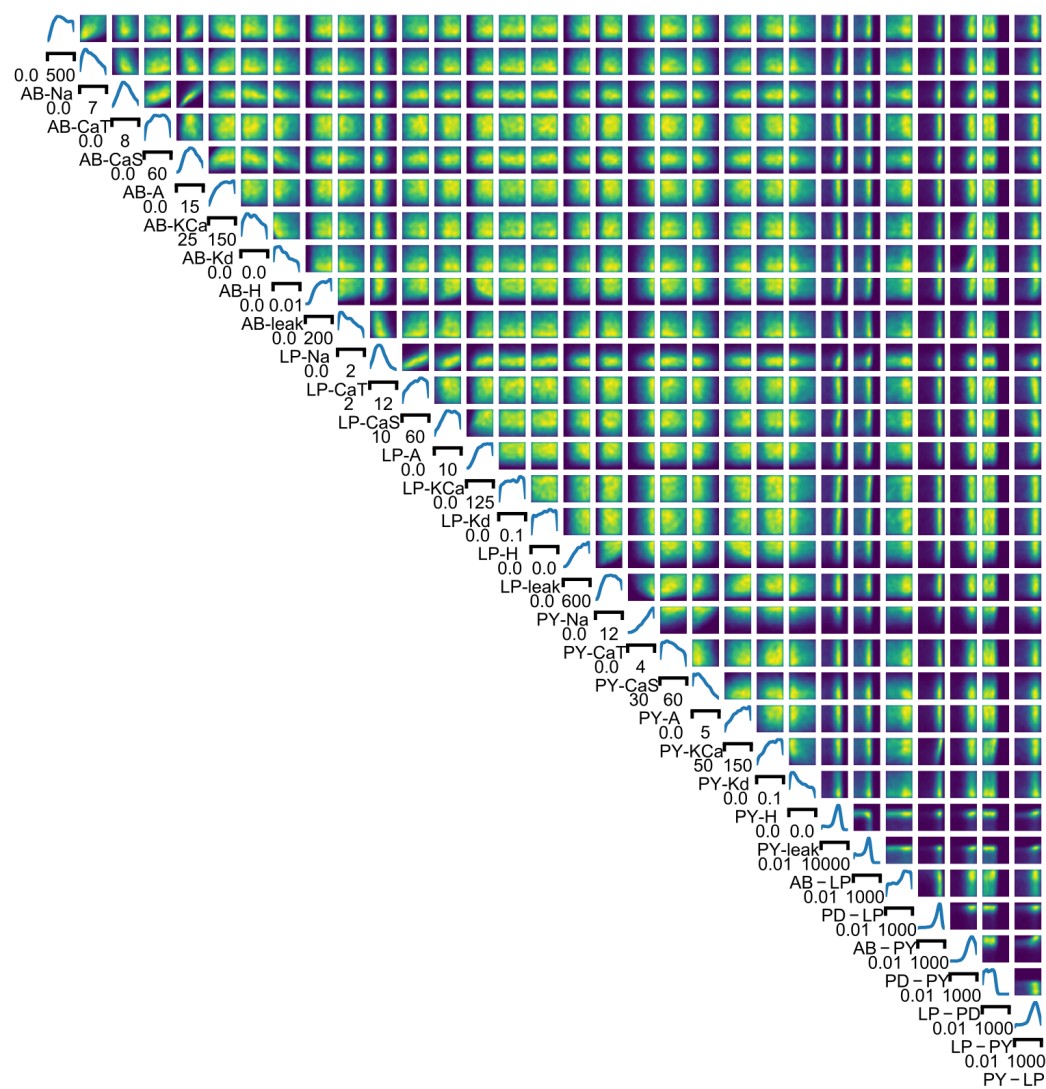

Figure 20: **Posterior distribution inferred by TSNPE for the pyloric network model.**

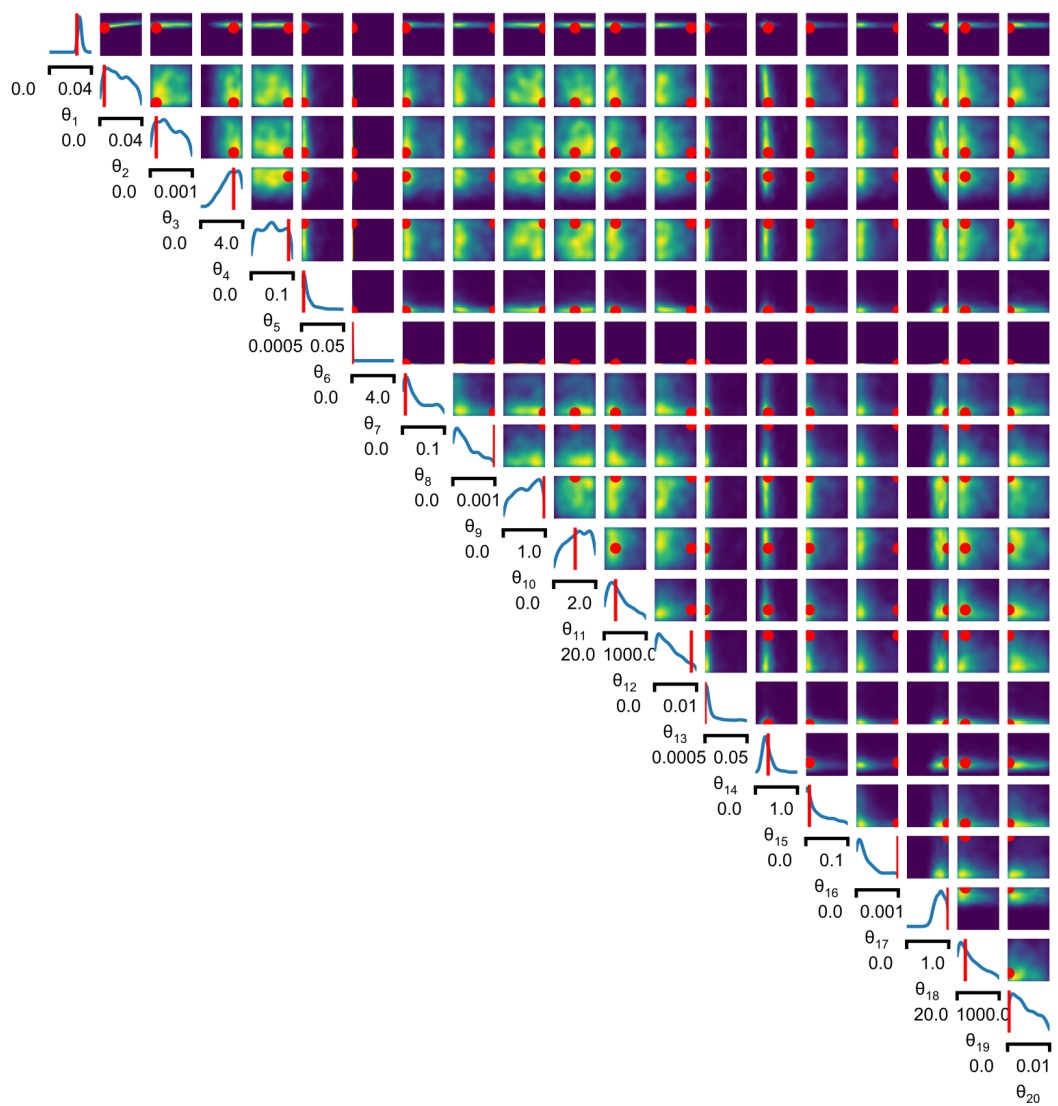

Figure 21: **Full posterior distribution over biophysical parameters for the layer 5 pyramidal cell.** Parameter names can be found in Appendix Sec. 6.14. Red dots are the ground-truth parameters.

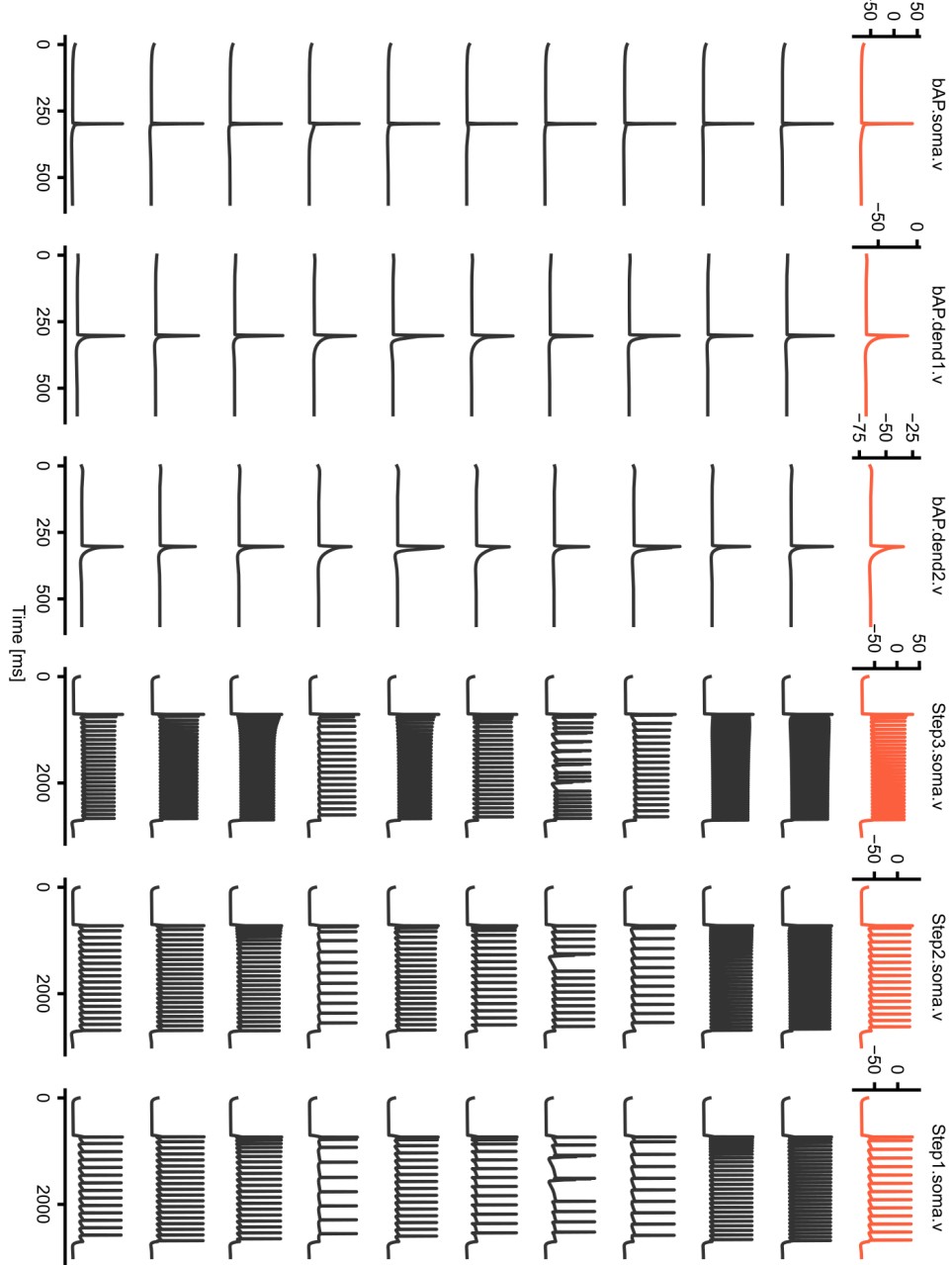

Figure 22: **Observed data (orange, top row) and ten posterior predictives obtained with TSNPE (black).**

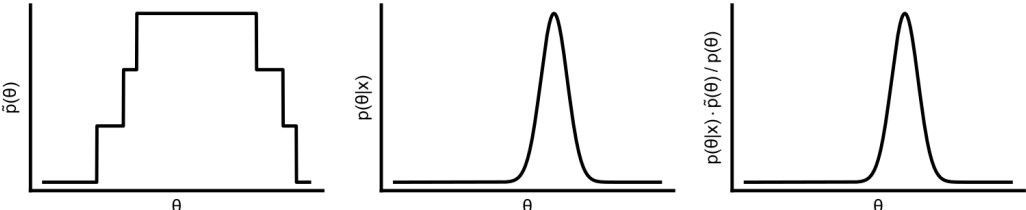

Figure 23: **Truncated proposals when pooling data from multiple rounds.** Assume a uniform prior. Left: Proposal that is the average of the proposal over three rounds. Middle: true posterior. Right: the approximate posterior converges to $p(\boldsymbol{\theta}|\mathbf{x}_o)\frac{\tilde{p}(\boldsymbol{\theta})}{p(\boldsymbol{\theta})}$ [Papamakarios and Murray, 2016, Lueckmann et al., 2017, Greenberg et al., 2019]. This matches the true posterior even though the proposal is not constant *outside* of the HPR$_\epsilon$ of the true posterior.