# OpenReview forum: "Truncated proposals for scalable and hassle-free simulation-based inference"
_NeurIPS.cc/2022/Conference — NeurIPS 2022 Accept_

### Official Review · Reviewer_Afw3 · 2022-06-24

**Rating:** 7
**Confidence:** 3
**Soundness:** 3 good
**Presentation:** 4 excellent
**Contribution:** 3 good

**Summary:**

The manuscript presents a new sequential neural posterior estimation (SNPE) algorithm for simulation-based inference. Simulation-based inference refers to the task of inferring parameters of interest from a real observation, based on a simulator. Neural posterior estimation (NPE) refers to methods aiming to build a surrogate for the posterior (typically a normalizing flow) based on samples from the simulator. Sequential NPE refers to their sequential counterparts that alternate between training and sampling phases with the objective of sampling simulated observations close to the real observation of interest and hence typically sample parameters to use for simulation from the current posterior surrogate. However, current SNPE methods suffer from two issues limiting their applicability:
* As training samples are sampled from the current posterior surrogate, they are non-amortized, meaning that the algorithms must be fully rerun for each observation. In opposition, amortized methods reuse the same model to perform inference on multiple observations. Non-amortized methods hence have the drawback of being inefficient to diagnose.
* As parameters used for simulation are not sampled from the prior, parameters with zero prior probability could be sampled. This phenomenon is called leakage.

Truncated SNPE (TSNPE) addresses both of those issues by using,  as proposal, a truncated version of the prior in place of the current surrogate posterior. The prior is truncated by removing parameters with surrogate posterior density below a given threshold, hence focusing on parameters of high posterior density. Consequently:
* Parameters sampled are always within the prior support, solving the leakage issue.
* It makes the method locally amortized as the built surrogate is valid for observations resulting from parameters sampled from a region of the prior instead of a single observation and hence enables efficient diagnostics. The paper introduces the simulation-based coverage calibration (SBCC) diagnostic that leverages this property.

**Questions:**

None

**Limitations:**

I do not see any potential negative societal impact of this work.

**Strengths And Weaknesses:**

### Major
* (+) The paper is very well written and easy to follow.
* (+) The method is sound
* (+) It addresses two important issues in SNPE (leakage and diagnostics)
* (+) Experiments are convincing.

### Minor

* (-) The SBCC diagnostic is quite known in the context of NPE and NRE (for example, see "Arbitrary Marginal Neural Ratio Estimation
for Simulation-based Inference", Rozet et al. 2021) and is a straightforward extension of SBC. Therefore, in my opinion, it should not be framed as a core contribution of the paper. The contribution rather lies in the development of an sbi algorithm that allows the use of this diagnostic (which is great).

---

> ### Author Response · Authors · 2022-08-02
> **Response to reviewer Afw3**
>
> We thank the reviewer for their insightful comments and feedback, as well as for appreciating our “well-written” paper and our “convincing” experimental results. We address the remaining concerns below and hope that this will allow the reviewer to give a strong recommendation for acceptance.
>
>
> > As parameters used for simulation are not sampled from the prior, parameters with zero prior probability could be sampled. This phenomenon is called leakage.
>
> We emphasise that this leakage can be a major limitation of APT (since leakage will often be 99.9999% of the posterior mass, as in the pyloric network example) and prevents its use-case on many real-world applications. TSNPE, thus, opens the possibility to apply simulation-based inference to models that have been inaccessible to APT and other SNPE methods.
>
> > The SBCC diagnostic is quite known in the context of NPE and NRE (for example, see "Arbitrary Marginal Neural Ratio Estimation for Simulation-based Inference", Rozet et al. 2021) and is a straightforward extension of SBC.
>
> We thank the reviewer for pointing us to this work, we had indeed not been aware of this. We have de-emphasized the diagnostic tool in the abstract and introduction, and re-written the corresponding section in the manuscript. As suggested by the reviewer, we highlight that TSNPE can make use of a diagnostic which scales to high-d parameter spaces and does not require MCMC sampling.

---

### Official Review · Reviewer_ue8u · 2022-07-07

**Rating:** 7
**Confidence:** 4
**Soundness:** 3 good
**Presentation:** 4 excellent
**Contribution:** 3 good

**Summary:**

The authors present a new simulation-based inference (SBI) technique that sequentially learns a neural surrogate for the posterior. The key idea is to use a truncated version of the prior, rather than the current posterior, as a proposal distribution to generate new samples. This removes the need for an impractical importance weight during training, leading to two practical advantages: it solves a posterior leakage problem that other state-of-the-art sequential neural posterior methods (APT) have, and it simplifies the use of calibration diagnostics. There are some downsides, too: the truncation biases the posterior and sampling from the proposal distribution requires some rejection sampling, which can be inefficient.

This new method, TSNPE, is demonstrated on a suite of standard benchmarks and finally on two real-life problems from neuroscience. On the benchmarks it performs on par with other neural posterior estimation methods, while on the neuroscience problems it enables plausible solutions for the posterior where other methods failed.

**Questions:**

The authors present the posterior leakage of APT as a major motivation behind their work. As they point out, this posterior leakage can be fixed in APT through rejection sampling, but that can be inefficient. However, the proposed algorithm (TSNPE) also involves rejection sampling (plain or in the form of SIR). I would appreciate if the authors discuss in more detail of why the rejection sampling in TSNPE is more efficient (has a lower rejection rate) than the rejection sampling in the APT posterior. Could SIR be applied equally well to the APT posterior?

Is it fair to say that in some sense, TSNPE is a compromise between vanilla NPE (no annoying importance factors in training) and SNPE (optimal use of past information in the posterior)? I'm trying to better understand how much information from past rounds is lost by sampling proportional to the prior rather than proportional to the current posterior (within some truncation region). Can the authors discuss this some more?

Phrasing the same question in a different way: Compared to standard SNPE, the proposed algorithm presents two changes. 1) The proposal is truncated and 2) within the truncated support, we sample proportional to the prior rather than the posterior. It would be interesting to disentangle these two factors and understand how they individually contribute to the sample efficiency. If I'm not mistaken, only doing 2 (and not 1) corresponds to vanilla, non-sequential NPE, is that correct? How would an algorithm do that does 1 but not 2, i.e. truncates the posterior and then samples proportional to the posterior within this truncation region? Would this for all practical purposes be identical to APT?

Some minor comments:
- The authors do a good job of listing the most closely related references. Perhaps SBI has grown to be too large a field by now, but there are still a number of SBI works not cited here.
- In the first sentence of the abstract, the authors describe SBI as "learning posterior distributions". As the authors themselves point out later, in many SBI methods other statistical objects like the likelihood or the likelihood ratio are learned, and other methods again do not have any explicit learning at all (like old-fashioned ABC). Maybe it is a good idea to rephrase this sentence.
- The word "likelihood" is a bit overloaded and used both for the likelihood $p(x|\theta)$, but often also to the density of some trained surrogate of the posterior $q(\theta|x)$. This is to some extent unavoidable, as "maximum likelihood" is such a known term, but in other instance I found it slightly confusing (for instance in line 128). Perhaps the authors could refer to the "probability density" of the posterior rather than the "likelihood" of the posterior?
- In the experiments, how big is the error of SIR due to finite K?
- In the benchmark experiments, does the problem of APT posterior leakage occur? Do you fix it with rejection sampling (or in some other way)?
- In the experiments, do the methods that rely on rejection sampling or SIR have a substantial overhead in runtime from that, or is that negligible?

**Limitations:**

The authors have addressed the limitations of their methods and discussed them well. Some aspects could use a more in-depth discussion, see the Questions section above.

The potential societal impact is not discussed, but I do not see any obvious issues.

**Strengths And Weaknesses:**

## Strengths

- Scaling SBI methods to more complex problem is an important (and to some extent unsolved) problem, with potential impact in many areas of science.
- The proposed method is, overall, sensible (with some caveats, see below) and can help make SBI work in more complex problems.
- The downsides of the presented methods (in particular the bias from $\epsilon > 0$ and the inefficiency of rejection sampling) are acknowledged and discussed (though this could be further improved, see below).
- The empirical evaluation is strong. I appreciate that the authors both present benchmark problems (for which true posterior samples are available) and more challenging real-world neuroscience problem. In the latter, the authors do a good job of discussing the plausibility of the learned posterior. While the presented method does not lead to a huge improvement on the benchmark tasks, it does appear effective in the neuroscience problems, and that is ultimately more important.
- The paper is very well written and clearly structured, quite a pleasure to read.
- The figures (both schematics and experimental results) are well designed and very clear.
- Related work is discussed in detail and the relation of the presented method to other approaches is made quite clear.
- The authors have released the code to run the experiments.


## Weaknesses

- While TSNPE is a new algorithm, it is a modification of existing algorithms like SNPE and APT. Similarly, the proposed calibration tool builds strongly on existing work. This paper is thus somewhat iterative rather than revolutionary.
- There are a couple of aspects that could use a more in-depth discussion, see the Questions section below. In particular, I would like to understand better why rejection sampling in TSNPE can be more efficient than rejection sampling based on the APT posterior. Also, it would be great if the authors could explain a bit more why sampling from a truncated prior isn't quite sample-inefficient compared to sampling from the posterior.

## Bottom line

I would like to thank the authors for a solid contribution to the toolbox of simulation-based inference and to a high-quality paper. Still, I think it can be made stronger by extending the discussion and addressing some questions (see below).

---

> ### Author Response · Authors · 2022-08-02
> **Reply to reviewer ue8u**
>
> We thank the reviewer for their insightful comments and for raising important discussion points. We have addressed all concerns and suggestions and believe that these changes have substantially improved our manuscript. We have added further analyses regarding (1) the impact of the information loss from using the truncated proposal rather than the posterior, (2) the error of SIR due to finite K, and (3) the computational cost of SIR and rejection sampling. We hope that these modifications clear remaining doubts of the reviewer and enable the reviewer to provide a strong recommendation for acceptance.
>
> > discuss why rejection sampling in TSNPE is more efficient than in APT.
>
> We thank the reviewer for raising this crucial point. The rejection rate of APT is not bounded: as we demonstrated on, e.g., the pyloric network application, the rejection rate can reach 99.9999% after only 2 rounds. In contrast, the rejection rate of TSNPE depends on the volume of the approximate posterior as compared to the prior. Across all benchmark tasks, the rejection rate was at most 99.9% and thus multiple orders of magnitude smaller than the rejection rate of APT. If the posterior is particularly narrow, we recommend using SIR in TSNPE, which requires K samples for each posterior sample (rejection rate 1-1/K). For K=1024, SIR has a rejection rate of 99.9%, which is again orders of magnitude more efficient than rejection sampling in APT. We discuss the computational cost of rejection sampling and SIR later in our response.
>
> > Could SIR be applied equally well to the APT posterior?
>
> This is an interesting idea, but in cases in which APT exhibits large leakage, SIR will not help. SIR is a resampling algorithm, which, in the case of APT, first draws K samples from the unbounded approximate posterior and then resamples one of them. However, if all K samples from the unbounded approximate posterior are outside of the prior bounds, the resampled sample will again be outside of the prior bounds.
>
> > how much information from past rounds is lost by sampling proportional to the prior rather than proportional to the current posterior
>
> We investigated this on the benchmark tasks and added the analysis to the paper: We compared APT with proposal being the truncated prior to APT with proposal being the previous posterior. Since both algorithms use the same loss function, it allows us to investigate the impact of the information loss, when using the truncated proposal as compared to the previous posterior, on the APT performance. We found that truncated APT shows similar or worse C2ST accuracy than standard APT but better performance than NPE, depending on the benchmark tasks (one task is still running). We note, however, that we cannot confidently say that this will also be the case on other models.
>
> > In the experiments, how big is the error of SIR due to finite K?
>
> We have added two analyses to the paper and now explore the efficiency of SIR in three ways: 1) As we show in Fig 10, SIR does not affect the posterior on benchmark problems. 2) We have added an analysis that demonstrates that, with K=1024, proposal samples follow the truncated proposal distribution almost perfectly on 1D toy examples (Fig. 11). 3) We studied the variance of the importance weights of SIR by inspecting the effective sample size (ESS). We found that, across the sir, 2-moons, and bernoulli-glm tasks, the worst ESS was 8.311, which is still far higher than 1 (the number of resampled samples). The other tasks are still running and we will update the manuscript accordingly. All of these results indicate that SIR is expected to be a useful method for TSNPE. We now address these points in a new supplementary section “Accuracy of SIR”.
>
> > In the benchmark experiments, does the problem of APT posterior leakage occur?
>
> On the benchmark experiments, leakage does not occur because the prior is transformed into unbounded space. This worked well on the low-dimensional benchmark tasks, but does not scale to high-dimensional problems (Fig 5).
>
> > In the experiments, do the methods that rely on rejection sampling or SIR have a substantial overhead in runtime from that?
>
> On a GeForce RTX 2080 GPU, one can draw (or evaluate) 100k samples from the approx. posterior in 0.17 seconds. When running SIR with K=1024, this produces 100 samples from the truncated proposal in 2x0.17=0.34 seconds. Thus, on most real-world simulators, the simulation time will be orders of magnitude higher than the SIR sampling time. E.g., for the multicompartment model (30 seconds per simulation), SIR sampling takes up only 0.012% of the time required to run the simulations. We added a new supplementary section “Computational cost of rejection sampling and SIR” to the paper.
>
> Further edits:
> - We have added several citations to the introduction.
> - We changed the abstract to “inferring posterior distributions from model-simulations”.
> - We changed several instances of the word “likelihood” to “posterior density”.

---

> > ### Comment · Reviewer_ue8u · 2022-08-05
> > **Thanks for the response and revision**
> >
> > Thanks to the authors for the updated paper version and the response, which I found very clear. In particular, I found the new experiments interesting and well-suited to address my questions.
> >
> > I've read the other reviews and the author's responses. Reviewer Afw3 made a good point about the novelty of the calibration diagnostic, but the authors addressed this well. The discussion about the similarities with TMNRE was interesting, but I share the authors' opinion that there are substantial differences between the methods and that TMNRE is sufficiently acknowledged and discussed.
> >
> > Overall, I think this paper presents a good contribution to the field and should be accepted. I am keeping my original score of 7.

---

> > > ### Author Response · Authors · 2022-08-08
> > > **Thank you!**
> > >
> > > We thank the reviewer for taking their time to review our paper and and for appreciating our efforts in the revision. We are happy that the reviewer can recommend our paper for acceptance.

---

### Official Review · Reviewer_VtwR · 2022-07-10

**Rating:** 7
**Confidence:** 4
**Soundness:** 4 excellent
**Presentation:** 4 excellent
**Contribution:** 3 good

**Summary:**

Given a generative model $\mathcal P(\theta, x)$ of the model parameters $\theta$ and data $x$, the goal is to solve the inverse problem $\mathcal P(x | \theta)$. Neural Posterior Estimation (NPE) uses simulations from $\mathcal P(\theta, x)$ to train an approximate posterior distribution $q_\phi (\theta | x)$ that is tractable and has learnable parameters $\phi$.
If one is interested in solving the inverse problem for a particular data realization $x_0$, Sequential Neural Posterior Estimation (SPNE) proposes a iterative sequence, starting at the prior, where the sampler for the model parameters is drawn from the previously learned $q_\phi$.

Rather than plugging in the previous proposal. this paper proposes a sequence of truncated prior distributions.
With Truncated Sequential Neural Posterior Estimation (TSPNE), the sampler for $\theta$ at each step is drawn from the prior distribution truncated at the highest-probability region (HPR) of the most recent approximate posterior.
This overcomes the drawback of SPNE that the maximizer of the loss function no longer converges to the true posterior.
The authors show that as long as the chosen HPR covers the support of the true posterior distribution, TSNPE provably converges to the true posterior.
The authors also introduce a metric called simulation-based coverage calculation (SBCC) that evaluates the coverage of their learned posterior approximation. This coverage is based on whether the quantiles of the learned log-density cover the log-density evaluated at the true parameter across repeated draws from the generative model.

The authors evaluate their procedure on six synthetic benchmarks and compare to the original NPE and APT, which is a specific variant of SPNE.
They show that their method mostly matches or outperforms APT and NPE, although APT seems to do better in two cases.
They then compare APT and their TSNPE on a simulation of pyloric networks in crabs.
Their TSNPE method does well in this example and is similar to previously known posteriors.
They also show that APT does poorly in this case.
Their final example is on a multicompartment model of a single neuron. Since only one proposal exhibited poor coverage based on their SBCC metric, they use an ensemble of 10 proposals and find this leads to much better coverage.
The competing APT method also does poorly in this case.


**Questions:**

+ Instead of an ensemble mixture to solve the poor coverage in the multicompartment experiment, would it work to form a defensive mixture by sampling from the untruncated prior with some probability? Even if this probability were small, it may help get around the issue of the HPR not covering the true posterior support.
+ It has been suggested that the forward KL objective (used here) leads to an overdispersed posterior compared to the reverse KL typical of variational inference. Is this applicable to TSNPE?


**Limitations:**

The authors address the main limitation of their method that the estimated HPR may leave out the true posterior support, and they describe their proposed SBCC metric for checking poor coverage.


**Strengths And Weaknesses:**

## Strengths
This paper builds on the established Sequential Neural Posterior Estimation (SNPE) framework, but using a truncated prior distribution as a prior. The truncation is achieved by rejecting samples outside of the highest-probability region (HPR) of the previously-estimated posterior. This method is simple, yet it makes intuitive sense because it prevents sampling parameters that are not possible under the prior and parameters unlikely under the estimated posterior.
Moreover, it corrects the main problem with SPNE by ensuring that the minimizer of the TSNPE objective at eatch stage is the true posterior.
This property hinges on the HPR covering the support of the true posterior, which may not be guarenteed in practice, but the authors develop the SBCC metric to verify the validity of the learned posteriors.
This lends confidence to the proposed method.

The pyloric network and multicompartment experiments demonstrate that TSNPE can work in challenging scenarios that the competing method (ANP) fails in. In the multicompartment experiment, it was nice to see that the authors' SBCC metric identified that a single neural network led to poor coverage and was corrected by an ensemble method.

## Weaknesses
The main weakness to this work are the simple simulation experiments in section 4.1. While TSNPE appears to match the performance of other methods, it would be reassuring to re-create the failure of APT observed in the more complex neurological applications in a simple setting.

---

> ### Author Response · Authors · 2022-08-02
> **Reply to reviewer VtwR**
>
> We thank the reviewer for their constructive feedback and for appreciating the strong results on real-world tasks as well as our efforts to identify and fix potential failure modes, which “lends confidence” to our method. Below, we address the remaining concerns of the reviewer:
>
> > it would be reassuring to re-create the failure of APT observed in the more complex neurological applications in a simple setting.
>
> We agree that showing this failure is important, and we demonstrate such a failure in Fig. 1. On a simple 1-dimensional toy example, APT leads to strong leakage. We have now added a supplementary figure which shows that this behaviour gets worse as even more rounds are run (see Appendix Fig. 7). This illustrates the instability of APT on a simple toy example and demonstrates why sampling in APT can become prohibitively slow.
>
> > would it work to form a defensive mixture by sampling from the untruncated prior with some probability?
>
> We thank the reviewer for this interesting suggestion. In this case, the proposal is no longer a truncated proposal, but a mixture between the prior and the truncated proposal. Because of this, we would no longer be able to train with maximum likelihood and would have to adjust the loss accordingly. This would lead to a mixture between the loss employed by Lueckmann et al. 2017 and our loss (i.e., maximum likelihood). We believe that exploring such mixtures between loss functions will be an interesting subject and potentially a fruitful idea for future research.
>
> > Is the forward KL objective applicable to TSNPE?
>
> Yes. Indeed, the loss-function of TSNPE is identical to minimizing the forward KL divergence between the true and approximate posterior and is, thus, support covering. We have highlighted this in the manuscript. This behaviour is highly desirable for TSNPE: since it relies on the proposal distribution being sufficiently broad, the mode covering behaviour of the forward KL confers a high degree of robustness to TSNPE.
> Summary: We thank the reviewer for their insightful comments and hope that our answers clarify any remaining doubts, so that they will hopefully be able to strongly endorse our manuscript.

---

> > ### Comment · Reviewer_VtwR · 2022-08-04
> > **Acknowledgement of author response**
> >
> > I thank the authors for their response. The toy model presented in Figure 1 is interesting and shows evidence of leakage. I appreciate the added figure in the Appendix of this toy model. After reading the authors' response to my review and the other reviewers, I keep my rating at a 7. After seeing Reviewer Afw3's review, I agree that the original submission oversold the novelty of their SBCC metric. However, the authors provided a good response by adding the relevant citations and editing the paper to reflect this.
> >
> > I disagree with Reviewer nSm9 that this method (TSPNE) is a simplifed version of TMNRE. TSPNE selects the truncation threshold of the joint distirbution based on the quantiles of the proposal density, while TMNRE truncates based on the ratio of the parameter to its maximum value. I find that the TSPNE threshold more intuitive, because it directly corresponds to the probability mass that is truncated at each stage of the algorithm, and it makes sense even as the dimension of the posterior grows. Moreover, it also seems that TMNRE was positioned to tackle a different setting than TSPME. It's unclear that TMNRE can handle a high-dimensional marginal posterior, since the highest marginal distribution they tested had $d=2$.

---

> > > ### Author Response · Authors · 2022-08-08
> > > **Thank you!**
> > >
> > > We sincerely thank the reviewer for taking their time to review our paper and are happy that the reviewer recommends our paper for acceptance!

---

### Official Review · Reviewer_nSm9 · 2022-07-10

**Rating:** 5
**Confidence:** 5
**Soundness:** 3 good
**Presentation:** 3 good
**Contribution:** 2 fair

**Summary:**

The paper introduces a simulation-based inference method for increasing the simulation-efficiency of Neural Posterior Estimation using an alternative proposal distribution. The proposal is constructed by truncating the prior to the HDR of the previously-estimated posterior for a particular observation of interest. They call their method Truncated Sequential Neural Posterior Estimation (TSNPE).

TSNPE:
- Overcomes issues with unstable training and leakage of posterior mass outside of the prior associated with other sequential version of NPE, namely so-called APT and SNPE-C.
- Enables a specialized coverage test, which is dubbed Simulation-based Coverage Calibration (SBCC), which exploits the tractability of the approximate posterior and general empirical expected coverage testing on the truncated region due to "local amortization."
- 's empirical performance is shown on six benchmark tasks from the simulation-based inference benchmark as well as two complex neuroscience problems. The performance on benchmarks is not significantly distinguished from other methods. The previous versions of Sequential Neural Posterior Estimation cannot robustly identify the posterior on the neuroscience problems.

**Questions:**

- Please address the bullet points raised in the "weaknesses" section above.
- Would you please clarify Algorithm 2 such that I can check whether or not it is novel?
- Would you please quantify the effects of using SIR and opposed to rejection sampling for drawing from the proposal? An analysis of the variance of importance weights would help us to understand whether SIR produces useful samples for training.
- Why do you consider computing coverage at every round to be optional? Truncating with an overconfident posterior, as is empirically produced by even the ensemble of networks in Figure 6, means that the proposal truncates too aggressively for the hyperparameter setting.
- I am not convinced that training TSNPE based on data aggregated from all rounds during truncation converges to the HDR of the ground truth posterior. Could you please clarify where in the paper this is proven or prove it here (and include it in the paper)? Specifically, I am concerned by how a collection of data from many rounds implies a prior with mass modulated by the details of truncation regions of previous rounds. It also implies that the estimated posterior outside of the HDR is untrustworthy.
- In Section 3.3 the authors claim that having (expected) coverage implies that the HDR of q_phi(t | x_0) is a super set of p(t | x_0). I don't think this is true. Would the authors please clarify or prove it?

**Limitations:**

- The simulation efficiency gained by truncation is marginal if the posterior is nearly as wide as the prior. This is mentioned in the result section, but should be put into the limitations section.
- It is possible to miss modes with this method and inaccurately remove them from the analysis. This should be emphasized. Coverage alone does not stop this from happening, to my knowledge. (see questions / quality weaknesses.)
- The limitations of computing coverage using the SBCC algorithms in high dimensions should be mentioned. Yes, drawing parameters is cheap from this method but it may be that a significant amount of proposed parameters are below the contour line of interest. I expect this to suffer from the curse of dimensionality.

**Strengths And Weaknesses:**

## Originality
### Strengths
- The combination of existing truncated proposals with existing Neural Posterior Estimation (NPE) is novel.
- Addressing the issue of posterior "leakage" into the region without prior support in APT / SNPE-C is an important advance in the development of NPE-related techniques.
- Although SIR is not novel, using it to sample from the truncated proposal in this context is new to me.

### Weaknesses
#### Relation to TMNRE
A core weakness of originality in the paper is that the proposed method is merely the combination of Neural Posterior Estimation with a simplified version (only rejection / SIR sampling from the prior; no iterative bounding of prior for faster proposal sampling) of the proposal algorithm of [Truncated Marginal Neural Ratio Estimation (TMNRE)](https://arxiv.org/abs/2107.01214), another deep-learning based and highly-effective simulation-based inference algorithm. This holds when the marginals are taken to be the entire set of parameters; a mode which is suggested in the TMNRE paper on page 6. Unfortunately, this current paper relegates the exceptional similarity with TMNRE to a paragraph which emphasizes that both techniques are locally amortized (agreed) and writes that their aims are orthogonal (a point on which I disagree). Furthermore, just as in the TMNRE paper, the current paper emphasizes empirical expected coverage testing in order to avoid truncating the prior too aggressively; however, the TMNRE paper is never cited in this context. To the reviewer's knowledge, the TMNRE paper is first to use tests of coverage in this way and even uses [a very similar algorithm](https://github.com/bkmi/tmnre/blob/main/tmnre/coverage/oned.py) for computing coverage as the current paper's proposed SBCC that doesn't require integrating on a grid as the current paper claims (consult their code, which is available on github). Finally, relevant to TMNRE, several examples of applications of simulation-based inference to scientific problems are cited; however, relevant work applying TMNRE (which uses the truncated proposals suggested by the current paper) is not cited. For example: [Fast and Credible Likelihood-Free Cosmology with Truncated Marginal Neural Ratio Estimation](https://arxiv.org/abs/2111.08030) and [Estimating the warm dark matter mass from strong lensing images with truncated marginal neural ratio estimation](https://arxiv.org/abs/2205.09126). Also the workshop paper which predated TMNRE, [Simulation-efficient marginal posterior estimation with swyft: stop wasting your precious time](https://arxiv.org/abs/2011.13951), is not cited either.

To address this issue, I suggest the authors:
- Clearly highlight the similarities between TMNRE and relationship to NPE. In particular, by emphasizing the similar methodology (in the case of parameters of interest = all parameters) applied to a different simulation-based inference technique. In my opinion, the connection is so close that it could be mentioned in the abstract along with the relationship to Blum and François [2010]. Also highlight the new key innovations contained in this paper.
- Clarify that the SBCC algorithm (as best as I can understand it given the state of its clarity in the current draft, see Clarity) is quite similar to techniques used in the computation of coverage by TMNRE and include TMNRE in citations in the discussions of using coverage as a diagnostic tool. (A very similar algorithm is also used in the averting a crisis in sbi paper, see github code.)
- Introduce the application / workshop paper citations suggested above.

#### SBCC Algorithms
The current paper suggests that coverage testing has only been done by integrating on a grid in the cited papers; however, this is not the case. An example is shown above in the previous section for TMNRE. Here is an example from the [Averting a crisis in SBI](https://github.com/montefiore-ai/averting-a-crisis-in-simulation-based-inference/blob/a53f34ccfa2b520c58f46158e52e235766da681f/workflows/SNL/mg1/snl.py#L116-L129) paper.. Both TMNRE and the crisis in SBI papers use a similar algorithm to SBCC and the crisis in SBI paper computes coverage for the entire set of parameters. In addition to the stated similarity to simulation-based calibration, this shows that the SBCC algorithm is not novel; although its explicit enumeration is useful for the community.

To address the issue:
- The paper should be updated to reflect that coverage is not generally computed by these other papers based on integrating on a grid.
- The paper should be updated to reflect that the SBCC algorithm is not novel.

**Caveat: Algorithm 2, which corresponds to SBCC has several typos. If this means that I didn't understand how it was novel, I would change my opinion here. Either way these typos must be fixed (see Clarity).**

#### Nested Sampling
The paper does not mention nested sampling, although truncation is an important part of the process. Please consider citing the following review https://arxiv.org/abs/2101.09675, and these papers:
- https://projecteuclid.org/journals/bayesian-analysis/volume-1/issue-4/Nested-sampling-for-general-Bayesian-computation/10.1214/06-BA127.full
- https://arxiv.org/abs/0809.3437
- https://arxiv.org/abs/1904.02180
- https://arxiv.org/abs/1506.00171


## Quality
### Strengths
- Addressing the leakage issue with APT / SNPE-C is demonstrated.
- The neuroscience tasks are made possible using the algorithm.
- The use of coverage testing to determine whether or not it is safe to truncate is a good idea and in line with previous work on the subject (TMNRE / crisis in sbi).

### Weaknesses
- The improvement on the benchmark is marginal, but the benchmark does not usually produce a clear "winner" among methods. Perhaps this should be stated within the paper.
- In Figure 6 d (and corresponding text) the use of ensembles does not produce a conservative estimator of the posterior, although it improves matters compared to single neural networks. Truncating in this round would lead to an overconfident estimate of the HDR. This is not mentioned in the paper, but it should be highlighted as exactly the failure mode to avoid!
- It should be made clear that computing coverage only in the final round does NOT tell the user if the algorithm truncated too aggressively since parameter samples are drawn from the already truncated prior. In this setting, it is like computing the coverage of different problem where the prior has been replaced by a truncated version.
- The term coverage is generally misused throughout the paper and not well explained in the proposed SBCC. The paper computes the expected coverage. Coverage is a frequentist concept which characterizes the confidence sets, not the credible regions discussed in this paper. Expected coverage is discussed the averting a crisis in sbi paper and implies a coverage test for averaged over simulations from the proposal. (See questions)
- Truncating is somehow "dangerous" when the first rounds of NPE miss a mode of the posterior, since it will not be represented in the subsequent rounds due to truncation. This is not sufficiently addressed and should have its own section. (See the limitations response prompt for more discussion.)
- It looks like the parameter rejection rate for many benchmark problems goes to nearly zero at higher rounds. The amount of help SIR provides in this is not well quantified; does using K=1024 distort the proposal distribution due to high variance in the importance weights? Does it affect the final posterior? This is an importance nuance which is not sufficiently explored. (See questions.)

## Clarity
### Strengths
- The main text is well written and easy to follow.
- I am extremely familiar with this field of research. I feel that it has a satisfactory preamble and main paper description. I will give a positive score but look to other reviewers to see if it is clear for others who are less well acquainted. Namely, I believe the paper references concepts from the development of deep-learning based simulation-based inference which are clear to me but perhaps not to others.
- I have not looked at the code presented in the paper, but hydra has been used for documenting the details of every run for reproducibility. That's very good!

### Weaknesses
- The proof of convergence in Section 6.2 is a difficult to follow. It could benefit from fewer words and more terse mathematical description.
- Algorithm 2 uses symbols which are starred then removes the stars later. It also uses symbols which are not introduced. This makes the algorithm unclear. Perhaps a verbal description would be beneficial as a comment? (See questions.)
- Algorithm 3 uses symbols which are not introduced. This makes the algorithm unclear. Perhaps a verbal description would be beneficial as a comment? (See questions.)
- The statement about the region which is well calibrated in 6.12 is unclear to me. Is the implication that the estimated posterior is truncated based upon the most recent round, thereby the support of q handles any issues regarding sampling parameters from the prior? If this is the case, it is a nuance which I suspect to pass most readers by and should be clarified. (see questions)
- It is not made clear that training on data drawn from all rounds should produce an accurate posterior. (Section 6.11) Drawing data from all rounds means that the "implicit prior," due to the use of a mixed proposal, can be modulated by a sum of R step functions with arbitrary support. Consider that in the limit of large steps that implies the modulating function can be extremely expressive. (see questions)

## Significance
### Strengths
- Although I am not an expert in neuroscience, the author's claims imply that the expanded relevance of NPE to these problems is significant.
- Addressing leakage in APT / SNPE-C is quite practical.

### Weaknesses
- As discussed in the Originality section above, the novelty of the TSNPE is marginal.
- Unless it is shown that training on data from all rounds is acceptable, then the proposed algorithm has a fundamental flaw. (It is addressed by training only on data drawn from that lies within the previous round's proposal, but then the performance on various tasks is unknown.)

---

> ### Author Response · Authors · 2022-08-02
> **Reply to reviewer nSm9, part 1**
>
> We thank the reviewer for their constructive and detailed feedback. We have updated our paper based on your suggestions, and believe that this has significantly strengthened our paper, and thus hope that the reviewer can now recommend the acceptance of the improved paper. We do, however, respectfully disagree with two of the central criticisms of the reviewer, namely the relationship of our algorithm to TMNRE and our coverage tests to those in previous work:
>
> Relation to TMNRE:
>
> We agree that our paper has close connections with TMNRE. We have further clarified this in the introduction and the corresponding paragraph in the discussion. In particular, we agree that we should have more strongly emphasised the contribution of TMNRE to using coverage as a diagnostic (see below).
>
> However, in our original submission, we cited and discussed the TMNRE paper—the reviewer is now requesting us to cite it in the abstract (!), to cite an additional workshop paper on it, as well as several papers applying TMNRE (all of which happen to have an overlapping set of authors). We are awaiting guidance from the AC as to which of these requests we should follow.
>
> The reviewer claims that “the proposed method is merely the combination of NPE with a simplified version [...] of the proposal algorithm of TMNRE”. We disagree, and believe that this statement does not do justice to several central contributions of our work:
>
> A) Importantly, TMNRE, in contrast to our algorithm, is focused on truncating marginals (hence the ‘M’ in its name), which implies a substantial difference in the algorithms in particular in higher-dimensional problems. The TMNRE paper focuses on one- and two-dimensional marginals—and only makes a single, passing reference on how NRE (not NPE!) could be generalized to truncating joints (on page 6, “In this paper, we estimate the one- and two-dimensional marginals necessary for corner plots. We emphasize that the data already generated during the truncation phase can be reused to learn arbitrary marginals of interest.”).
>
> Other than this passing reference, if and how TMNRE would work in this mode was never investigated in the paper. Several key questions on how to use TMNRE to truncate the full posterior are not addressed in the paper, for example: is the value of the density threshold epsilon=1e-6 still justified in high-D parameter spaces? How well does nested sampling, as proposed by TMNRE, perform in high-d spaces (since TMNRE requires rejection steps, it might be substantially slow)? How can one extend their diagnostic tool to high-D spaces (see below)?
>
> There can be big differences between looking at joint posteriors or marginals. If we had based our truncated proposals on the one-d marginals only, gains in simulation efficiency would have been orders of magnitude smaller: in the pyloric network model, truncation based on the marginals reduces the prior volume to 80% of the full prior (the marginals cover almost the full prior range), whereas truncation based on the joint reduces the prior volume to 0.06%. In other words, truncation based on the joint provides a speed-up of several orders of magnitude compared to truncation based on the marginals. We have added a discussion paragraph that more clearly describes the shortcomings of TMNRE and describes the main solutions provided by TSNPE.
>
> In summary, while we agree that TMNRE and TSNPE share several features (and we have further emphasised this in the introduction and discussion), our work provides key innovations that had not been addressed in TMNRE. We have added further details to more clearly describe both the similarities, but also the major differences between the algorithms.
>
> B) Notably, NRE and NPE are distinct approaches, and they have complementary strengths and weaknesses—therefore, developing a truncated approach for TSNPE is important and a useful contribution. We demonstrate (and prove) that the use of truncated proposals allows us to perform SNPE without correction of the loss function, thus side-stepping problems of previous SNPE methods. The fact that one can use the loss function of NPE (instead of the atomic loss of APT) is a central contribution of our work and allows us to scale SNPE to new and impactful real-world inference problems, which have previously been inaccessible to any SBI approach. We have now improved our convergence proof which hopefully makes it clear how TSNPE works, and shows that the combination of truncated proposals with NPE is non-trivial and that solutions to these problems were not a subject in the TMNRE paper.
>
> We address concerns regarding clarity and correctness of our method later in our response. We are confident that our approach does not contain a ‘fundamental flaw’.

---

> > ### Author Response · Authors · 2022-08-02
> > **Reply to reviewer nSm9, part 2**
> >
> > Relation of coverage tests to previous work
> >
> > The reviewer writes that the diagnostic tools used in TMNRE and in the averting-a-crisis-paper do not use a grid. We respectfully disagree with the reviewer on this point. We checked the code of TMNRE on github and found that it uses a histogram of samples for computing the coverage [here](https://github.com/bkmi/tmnre/blob/40d2bf2faf037a211923ce22130913ff996e88b7/tmnre/coverage/oned.py#L66). Such histogram-based approaches work well in 1D parameter spaces (as is done in TMNRE for the 1D marginals), but do not scale to high-dimensional parameter spaces. This highlights that TMNRE cannot easily be extended to diagnosing coverage of the full posterior (it would require expensive MCMC runs for multiple observations. The performance of this was never empirically investigated in the TMNRE paper). Similarly, the code of the averting-a-crisis paper [here](https://github.com/montefiore-ai/averting-a-crisis-in-simulation-based-inference/blob/a53f34ccfa2b520c58f46158e52e235766da681f/workflows/SNL/mg1/snl.py#L94-L98) shows that coverage is based on evaluating the posterior distribution on a grid which, again, does not scale to high-D parameters spaces (all examples they show are one, two, or three-dimensional).
> >
> > The reviewer also claims that they “expect this [SBCC] to suffer from the curse of dimensionality”. However, this is not true. Independent of the dimensionality of the problem, the fraction of “proposed samples” (i.e., posterior samples) that are below the contour line is exactly the empirical coverage. Thus, SBCC does not suffer from the curse of dimensionality in the same way as histogram-based approaches (e.g., TMNRE) or grid-based approaches (e.g., crisis-paper) and, therefore, constitutes a clear advantage of TSNPE. We note that one could, in principle, evaluate the coverage of the full posterior also for likelihood(-ratio)-based approaches such as TMNRE without evaluating on a grid, but this would require MCMC sampling from the posterior given several observations, which is substantially slow. We now discuss this more clearly in the paper.
> >
> > Overall, we maintain that neither the coverage-computation of TMNRE nor that of the crisis-paper provide any evidence that they can be evaluated quickly in high-dimensional parameter spaces, in contrast with SBCC. We have clarified these shortcomings on TMNRE and the crisis-paper in the manuscript and highlighted that TSNPE allows coverage computation of the joint posterior without evaluating on a grid or MCMC sampling.
> >
> > Further discussion points:
> > > In Figure 6 d [...] Truncating in this round would lead to an overconfident estimate of the HDR.
> >
> > We agree that the posterior is rather underconfident. However, for truncation, the crucial regions of the coverage tests are for high-confidence regions (because these define the HPR for small epsilon and, thus, undercoverage in these regions would indicate that parameters are wrongly excluded from the support). In these regions, coverage is very good and we, therefore, expect a reasonably good estimate of the HPR. Nonetheless, we have emphasised this danger in the results section.
> >
> > > It should be made clear that computing coverage only in the final round does NOT tell the user if the algorithm truncated too aggressively since parameter samples are drawn from the already truncated prior.
> >
> > We absolutely agree on this point. As spelled out in Algorithm 1, we compute coverage at every round.
> >
> > > Truncating is somehow "dangerous" when the first rounds of NPE miss a mode of the posterior, since it will not be represented in the subsequent rounds due to truncation.
> >
> > We agree that this could be a problem and we have highlighted this in the discussion. We do, however, emphasise that the loss of NPE tends to be mode-covering (it is the forward KL). In addition, it is expected that sequential methods that do not rely on truncation will suffer from these issues as well.
> >
> > >  It is possible to miss modes with this method and inaccurately remove them from the analysis. [...]. Coverage alone does not stop this from happening, to my knowledge.
> >
> > As discussed above, we agree that it is important to cover all modes. However, if the approximate posterior consistently misses one mode (for several observations), this would be reflected in the coverage diagnostic. The diagnostic is limited by the fact that SBC(C) only evaluates the coverage on average and does not make statements for particular observations. We have emphasised this problem of SBC(C) in the manuscript.

---

> > > ### Author Response · Authors · 2022-08-02
> > > **Reply to reviewer nSm9, part 3**
> > >
> > > > Regarding rejection rate and SIR: Does it [SIR] affect the final posterior?
> > >
> > > We explore the efficiency of SIR in three ways: 1) As we show in appendix Fig 10, SIR does not affect the posterior on benchmark problems. 2) We have added an analysis that demonstrates that, with K=1024, proposal samples follow the truncated proposal distribution almost perfectly on 1D toy-examples (Appendix Fig. 11). 3) As suggested by the reviewer, we have studied the variance of the importance weights of SIR by inspecting the effective sample size (ESS). We found that, across the sir, two-moons, and bernoulli-glm tasks, the worst ESS we observed (for K=1024) was 8.311, i.e., it was always significantly higher than 1 (the number of resampled samples). The results for other benchmark tasks are still running and we will update the manuscript accordingly. All of these results indicate that SIR is expected to be a useful and robust sampling method for TSNPE. We now address these points in a dedicated supplementary section “Accuracy of SIR”.
> > >
> > > > The proof of convergence in Section 6.2 is a difficult to follow.
> > >
> > > We thank the reviewer for pointing this out. We have clarified the proof and made it more rigorous for data pooled from all rounds. We hope that this clarifies all doubts about the correctness of our method.
> > >
> > > > The statement about the region which is well calibrated in 6.12 is unclear to me.
> > >
> > > We thank the reviewer for pointing this out. We have rewritten this paragraph in order to make it more understandable. In short, the issue arises because our proof only ensures convergence to the true posterior for the observation x_o, but not for other x. This poses a difficulty to applying SBCC (which computes the posterior for several x). We now describe two ways in which this issue can be circumvented.
> > >
> > > In Section 3.3 the authors claim that having (expected) coverage implies that the HDR of q_phi(t | x_0) is a super set of p(t | x_0).
> > > We agree with the reviewer that this is inaccurate. As with all SBC methods, the expected coverage only implies that the HPR is correct on average, but not for a particular observation. We have corrected this in the manuscript.
> > >
> > > >The simulation efficiency gained by truncation is marginal if the posterior is nearly as wide as the prior.
> > >
> > > We thank the reviewer for raising this point. We agree that, if the posterior is as wide as the prior, our method will not be more efficient than NPE. However, this is also expected for other sequential methods such as SNPE (or SNLE, SNRE, TMNRE). Thus, we do not believe that it is necessary to further elaborate upon this point beyond mentioning it in the results section.
> > >
> > > > The limitations of computing coverage using the SBCC algorithms in high dimensions should be mentioned.
> > >
> > > As we have argued above, this is not correct. Unlike the coverage methods suggested in TMNRE and the averting-a-crisis paper, SBCC scales well with dimensionality.
> > >
> > > Further modifications:
> > > - We thank the reviewer for pointing out the typo in algorithm 2. We have updated its description.
> > > - We have added nested sampling as a possibility to sample from the truncated proposal.
> > > - We have clarified algorithm 3 with comments.
> > > - We have updated the term coverage to expected coverage.
> > > - We have removed the word “optional” from the computation of coverage as a part of TSNPE.

---

> > > > ### Comment · Reviewer_nSm9 · 2022-08-07
> > > > **reply 3**
> > > >
> > > > tThanks for considering these points and making changes. Generally there isn't much to say on these topics. I think that the proof in 6.2 is clearer now, but I still wonder about its fundamental assumptions as I stated above. It would be great to hear about this from you.
> > > >
> > > > Thank you for taking the time to make these changes. In general the paper is obviously well written and I think it does a fairer job at citing exciting literature now. My concern lies primarily in the assumption I mentioned above. If this can be properly addressed I would certainly recommend acceptance. If not, I think it's important to reevaluate where the results of this algorithm are applicable. (my intuition would be only in the truncated region of the prior.)
> > > >
> > > > Thank you!

---

> > > > > ### Author Response · Authors · 2022-08-08
> > > > > **Response to comments by reviewer nSm9**
> > > > >
> > > > > We thank the reviewer for taking their time to respond and for the constructive feedback. We briefly address a few minor points and then discuss the assumption made in our proof of convergence. We hope that our response, as well as our new figure and changes to the manuscript, convince the reviewer to recommend our paper for acceptance.
> > > > >
> > > > > > In TMNRE the value of truncating the marginals is to make sampling from the prior simple
> > > > >
> > > > > We agree that the fact that TMNRE proposals allow for quick sampling is a nice feature of these proposals. We have added this to the discussion:
> > > > >
> > > > > ```truncating based on the marginals allows TMRNE to sample from the truncated proposal without rejection or SIR sampling.```
> > > > >
> > > > > > The critique about rejection steps in TMNRE confuses me.
> > > > >
> > > > > The “critique about rejection steps” referred to the case in which the truncation is performed on the joint posterior and sampled with nested sampling. This is mentioned in the second to last paragraph in the discussion of the TMNRE paper. We are sorry about any confusion regarding this point. We agree that truncation based on the marginals does not require rejection sampling.
> > > > >
> > > > > > That's an issue with scalability.
> > > > >
> > > > > As we write in the manuscript, we agree that rejection sampling is limited if the rejection rate is too high. However, even rejection rates of 99.9999% require significantly less computational cost for sampling than for running simulations (for many real-world simulators): As we discuss in Sec. 6.11, the cost of drawing 100 rejection samples at a rejection rate of 99.9% is only 0.012% of compute time as compared to running simulations. Thus, even for rejection rates of 99.9999%, sampling is significantly cheaper than simulating. In cases in which the rejection rate becomes prohibitive, we recommend using SIR.
> > > > >
> > > > > Finally, we note that our example with highest dimensionality is 31D (Fig 5), not 20D. We emphasise that most problems tackled in sbi literature have less than 10 parameters, and we are not aware of any work that scaled sequential neural-network based SBI to a simulator with more than 31 parameters. Thus, our results suggest that TSNPE is at least as scalable as other sequential SBI methods in problems with 31 dimensions or less.
> > > > >
> > > > > > If this assumption does not hold, then I think the proof does not either.
> > > > >
> > > > > We thank the reviewer for clarifying their concerns about convergence of our algorithm. We agree that, if the assumption does not hold (i.e., if the truncated proposal at any round does not cover the posterior support), then the proof does not hold. As we had written in the original submission:
> > > > >
> > > > > ```Values of ε > 0 yield a proposal prior which has smaller support than the current estimate of the posterior, e.g., using ε=1e-3 neglects 0.1% of mass from the approximate-posterior support. Thus, this approach can lead to errors in posterior estimation, e.g., to `under-covered' posteriors.```
> > > > >
> > > > > We agree that we had not explicitly spelled out how this failure affects inference performance, and neither had we demonstrated this failure on a toy example. To address this, we have added a new supplementary section “Approximation errors due to truncation” in which we describe this failure mode in detail and demonstrate it on a 1D toy example (Fig. 11). We will move parts of this section into a dedicated “Limitations” paragraph in the discussion for the camera-ready version (as an extra page is available).
> > > > >
> > > > > On benchmark and real-world problems, however, we do not expect the truncation errors to strongly influence inference quality: We extensively benchmarked different truncation thresholds ε > 0 (Fig 4, Appendix Fig 8 & 9) and across all benchmark tasks, we found that the exact choice of threshold does not significantly influence C2ST accuracy. This indicates that the error of our method (as well as other (S)NPE methods) is dominated by limited simulation-budgets and imperfect neural network convergence.
> > > > > Overall, while we agree that truncation might lead to errors in the tails of the posterior approximation (and we explicitly highlight this in the new supplementary section), our theoretical and empirical results demonstrate that the errors (with ε < 1e-3) are expected to only weakly affect inference quality.
> > > > >
> > > > >
> > > > > > The overconfidence exhibited by some of your plots makes me wonder if this is a symptom of the issue I raised in part B. [...] Isn't that what we see in Figure 6d?
> > > > >
> > > > > The coverage plots shown in Figure 6d are after the first round (i.e., no truncation yet). Thus, truncation cannot be the reason for the observed overconfidence in this example. In general, while we agree that the errors induced by truncation should, in principle, be visible in the high-confidence regions of SBCC, the errors induced by ε=1e-4 are very small: Nominal coverage of 1 would correspond to 1-ε empirical coverage, which is 0.9999 for our choice of ε. Similar to what we wrote above, we expect errors due to limited simulation-budgets to dominate the errors observed in SBCC.

---

> > > > > > ### Author Response · Authors · 2022-08-09
> > > > > > **Awaiting reviewer response**
> > > > > >
> > > > > > Given that the discussion period is about to end, we would be happy if the reviewer informed us whether our response convinces them of the soundness of our method, or, if not, what further evidence (in addition to the newly added figure and section in the manuscript) would be required to change their mind. We again thank the reviewer for taking the time to review our paper, we believe that their feedback has significantly strengthened our manuscript!

---

> > > > > > ### Comment · Reviewer_nSm9 · 2022-08-09
> > > > > > **good reply**
> > > > > >
> > > > > > I will reply primarily to my concern about the assumption. I think your numerical and empirical arguments are strong and go along with how truncation has been justified in existing literature. I'm glad that you clarified when the assumptions can hold and also I should have read more closely to see the epsilon discussion that you pointed out.
> > > > > >
> > > > > > Generally, I think you have put in a significant amount of work addressing my concerns which is appreciated. Also I think that the main points about the nature of the truncation are well explained. I will change my ratings to reflect this and recommend acceptance.
> > > > > >
> > > > > > Congrats!

---

> > > > > > > ### Author Response · Authors · 2022-08-09
> > > > > > > **Thank you**
> > > > > > >
> > > > > > > We thank the reviewer for their immense efforts in reviewing our paper and are happy that the reviewer can now recommend acceptance!

---

> > > ### Comment · Reviewer_nSm9 · 2022-08-07
> > > **reply 2**
> > >
> > > After looking at the examples, I think that you are correct. Those methods do indeed evaluate on a grid. I was wrong.
> > >
> > > I appreciate that you updated your algorithm on this matter. I disagree that doing MCMC with a ratio estimator is a fundamentally difficult or slow task, but in general I think you clarified your point well here.
> > >
> > > I'll skip to the last paragraph about coverage. The overconfidence exhibited by some of your plots makes me wonder if this is a symptom of the issue I raised in part B. If indeed you are using a mixture of proposal distributions which are not accounted for in the loss, I would expect that the posterior is overconfident in the tails. (tails are too shallow.) Isn't that what we see in Figure 6d?

---

> > ### Comment · Reviewer_nSm9 · 2022-08-07
> > **reply 1**
> >
> >  Dear authors,
> > Thank you for your thoughtful reply! It's clear you spent some time considering my thoughts and I appreciate it.
> >
> > I spoke with the AC and we agreed that it was too much to ask to cite other works in the abstract. The changes made in the new version and citations are satisfactory to me. Although, I would request that you consider the following reply when characterizing TMNRE in the discussion.
> >
> > A) Indeed, in TMNRE the selection of the truncated prior and the computation of the "final" estimates are separated. The initial phase in TMNRE generates samples relevant to a certain observation and the second phase is to estimate arbitrary posterior marginals (including the joint) on the data generated in this truncated region. By truncating marginals, the data is generated in a "product space" of truncated priors and may take up a more prior volume than by truncating the joint. The authors are correct that samples from this space are likely to be LESS simulation efficient (posterior accuracy / simulation count) than the ones drawn using their proposed method of truncating the joint; however, the proposed method for drawing from the joint still raises some concerns for me both in novelty and scalability.
> >
> > I claimed that TSNPE is a simpler proposal mechanism that TMNRE and I still believe this. The reason is that both of the experimentally investigated methods to draw from the truncated prior in TSNPE, rejection sampling and SIR, are completely applicable to TMNRE applied to a truncated joint. The critique about rejection steps in TMNRE confuses me because the whole point of TMNRE is to avoid rejection to sample from the prior. In TMNRE the value of truncating the marginals is to make sampling from the prior simple. This is necessary when the parameter space is high dimensional. It merely involves the integral probability transform and changing the bounds of the interval.
> >
> > TSNPE does explore doing truncation in higher dimension than TMNRE, but the experiments in TSNPE have maximally 10-dimensional parameters in the toy problems and 20-dimensional parameters in the scientific application case. As seen in Figures 7-9 in the appendix, the acceptance rate of drawing from the truncated prior goes to nearly zero in TSNPE. That's an issue with scalability.
> >
> > Given these points, I don't really see any "major innovation" in truncation methods in TSNPE.
> >
> > B) The use of the "standard" loss function in NPE without correction is, in my opinion, the primary advantage and notable point (innovation) of this paper. This is also what I claimed in my initial review. Correcting for the proposal distribution is a major problem in SNPE as it really produces overconfident posteriors.
> >
> > I was not careful with my words. "Fundamental flaw" is too harsh, especially without proper explanation. For that I apologize. Let me be specific now, I am concerned that in appendix 6.2 your proof assumption that the HDR is not necessarily the support of the posterior. Consider a gaussian model with a conjugate likelihood. The support of the posterior distribution is R but any HDR is a compact set.
> >
> > Why did I say this is a fundamental flaw? If this assumption does not hold, then I think the proof does not either. I think it is likely that you are using a proposal distribution which needs to be accounted for in the estimate of the posterior, if you want the posterior to be accurate everywhere. I think this glosses over a really critical point of truncation: you cannot trust the estimate, except perhaps on the truncated region.

---

### Author Response · Authors · 2022-08-02
**General reply to reviewers**

We thank the reviewers for their extensive comments and insightful feedback on our manuscript. The reviewers highlight the relevance of our work, call our work an “important” (nSm9 & Afw3) advance in SNPE methods and appreciate that our work might have “impact in many areas of science” (ue8u). They also appreciate our “convincing” experiments (Afw3) and our efforts to diagnose possible failure modes of our method (which “lends confidence to the proposed method” (VtwR)). Finally, we are happy that the reviewers believe our paper to be “very well written” (Afw3), “quite a pleasure to read” (ue8u) and a “high-quality paper” (ue8u). We thank the reviewers for this highly positive feedback.

In addition to requests for clarification, the reviewers suggest a number of experiments to further strengthen our paper. To address this, we have added supplementary figures that:
1) compare inference quality of APT with and without truncated proposals;
2) show that SIR sampling has a sufficiently high effective sample size (ESS) to produce good samples from the truncated proposal;
3) demonstrate that the computational overhead of rejection sampling and SIR is small compared to the cost of simulating.

Finally, as suggested by reviewer nSm9, we also more clearly explain how our submission is substantially different from previous work (in particular “Truncated marginal neural ratio estimation”, TMNRE).

We hope that our responses address the remaining concerns of all reviewers and that they can recommend our paper for acceptance at NeurIPS. We thank the reviewers for taking their time to review our paper.

---

### Comment · Area_Chair_N1ox · 2022-08-04
**Some concerns with SIR sampler**

Dear Authors

Thank you very much for your submission and extensive efforts in replying to the reviews.

I have just gone through the paper myself and wanted to raise a separate concern I had to those mentioned by the reviewers.  To avoid giving you a heart attack, I should point out upfront that it is not something I think is a fatal flaw, but I am trying to gather as much information as possible for making the final decision and was hoping you might be able to provide some comments on it.

> SIR seems to me to be a problematic and potentially unstable mechanism for sampling from the truncated prior.  I’m sure that it will often work absolutely fine, but it introduces an unwanted interplay between the fact that the NPE at each iteration is used to define the truncated prior estimator that will be used to update it in the next.  It would thus be quite easy to have an initial underestimate of the posterior density in some region reinforce itself in a way that leads to a region of the true posterior being missed.  This could be particularly problematic when we consider the fact that the TSNPE objective itself requires the approximation to be more disperse than the true posterior.  From a more theoretical perspective, the SIR estimator is biased with a bias that depends on phi itself (and which is not guaranteed to diminish with K if the tails of q are too light). This will mean that it is difficult to prove any convergence for the approach; indeed, I would expect it to sometimes diverge.  I think this needs to be properly acknowledged as a potential limitation and I would encourage the authors to consider testing alternative, less problematic, sampling schemes as well, with nested sampling, adaptive multi-level splitting, or even resample-move SMC potential suitable candidates for this.  Note that the instabilities I expect to see with SIR here are quite common in the adaptive importance sampling literature and the approach is kind of working like an adaptive importance sampler in that current approximations are driving the sampling in an adaptive way.  This literature might thus be helpful for categorising the potential issues that might occur.

As I said, it would be good to hear your thoughts on this and, if you do think the concern is reasonable, whether there is anything you might be able to do to address/alleviate it.

Thanks

AC

---

> ### Author Response · Authors · 2022-08-08
> **Reply to AC**
>
> We thank the AC for their interest in our work and for this insightful comment! We have added a new paragraph to the paper and we have added a supplementary figure in which we demonstrate that, indeed, SIR with too low values of K leads to a too narrow proposal distribution and, thus, a too narrow TSNPE posterior approximation (new supplementary Fig. 12). When run for a large number of rounds, this behaviour reinforces itself and can lead to divergence of TSNPE. In our benchmark and real-world experiments, we did not observe this behaviour (with K=1024 and 10 rounds), but we now specifically emphasise the importance of using tools to diagnose potential failures of TSNPE (e.g. SBCC) in combination with diagnostic tools for SIR (e.g. by inspecting the importance weights as we suggest in Supplementary Sec. 6.12). Finally, we agree that, in cases in which SIR fails (and rejection sampling cannot be applied due to its high rejection rate), sequential methods such as the ones mentioned by the AC provide a viable option to sample from the truncated proposal.

---

> > ### Comment · Area_Chair_N1ox · 2022-08-08
> > **Thanks**
> >
> > Thanks for your reply, this is indeed helpful and I think the changes improve the paper.  If the paper is accepted, I think it would be good to use the extra page available for the camera-ready to build on these a little further and the potential failure mechanism more directly in the main paper, but I appreciate there is limited space at the moment and there is no need to do this now.

---

> > > ### Author Response · Authors · 2022-08-09
> > > **Thank you!**
> > >
> > > We are happy that the AC finds our changes helpful. We agree that further spelling this out in the main paper will be useful for practitioners and we will do so if the extra page becomes available. Thank you for taking your time to assess our paper!

---

### Meta-Review · Area_Chair_N1ox · 2022-08-30

**Recommendation:** Accept
**Confidence:** Certain

**Metareview:**

This paper received generally positive reviews, with one reviewer originally weakly backing rejection but ultimately backing acceptance after substantial discussions, and the other three confidently backing acceptance from the off.  Based on the reviewer's comments and my own assessments, I see the positives and negatives of the work as follows:

Positives
- Very strong writing and presentation
- Good range of experiments
- Key ideas seem relatively simple to use and deploy
- Important problem area and seems to make decent progress on two known issues in the area (leakage and coverage tests)

Negatives
- The novelty of the work is quite limited: the core approach is arguably a special case of an approach that already exists (APT) and the paper is mostly combining known ideas rather than proposing anything especially original. Thus, though the paper does certainly have some new ideas and is potentially useful to the community, I do think it is quite incremental.
- The use of SIR is worrisome and likely to cause theoretical and occasional practical failure cases, even if these have not especially manifested in the current experiments
- The method introduces biases (from SIR and the truncation itself) that might be difficult to quantify and there is a lack of any notable theoretical guarantees (though the objective is quite clearly sound with epsilon=0 and no iteration of the truncation, this provides no guarantees for real-world settings that will have epsilon>0 and multiple rounds of truncation).
- Some of the claims about efficiency and rejection rates could have been more clearly demonstrated and explained.

On balance, I agree with the reviewers that the positives outweigh the negatives; this is well-polished and potential useful work that will be of interest to the community.  As such, I recommend acceptance.

Minor comment
- I believe that the suggestion that TSNPE “can scale to problems that were previously inaccessible to neural posterior estimation” is over claiming given the actual experiment results, given the fact that only one approach (with a single “out of the box” proposal) is actually compared to and the fact that the results are generally qualitative rather than quantitative.  Moreover, given that TSNPE can arguably be seen as a special case of APT in the first place, the claim seems distinctly unreasonable.  I would thus like to see this claimed dropped or at least significantly toned down.

**Award:**

No

---

### Decision · Program_Chairs · 2022-09-14

Accept